# Gradient-based Discrete Sampling with Automatic Cyclical Scheduling

**Patrick Pynadath**
Department of Computer Science
Purdue University
West Lafayette, IN, 47907
ppynadat@purdue.edu.edu

**Riddhiman Bhattacharya**
Department of Management
Purdue University
West Lafayette, IN, 47907
bhatta76@purdue.edu

**Arun Hariharan**
Department of Computer Science
Purdue University
West Lafayette, IN, 47907
harihar4@purdue.edu

**Ruqi Zhang**
Department of Computer Science
Purdue University
West Lafayette, IN, 47907
ruqiz@purdue.edu

## Abstract

Discrete distributions, particularly in high-dimensional deep models, are often highly multimodal due to inherent discontinuities. While gradient-based discrete sampling has proven effective, it is susceptible to becoming trapped in local modes due to the gradient information. To tackle this challenge, we propose an automatic cyclical scheduling, designed for efficient and accurate sampling in multimodal discrete distributions. Our method contains three key components: (1) a cyclical step size schedule where large steps discover new modes and small steps exploit each mode; (2) a cyclical balancing schedule, ensuring "balanced" proposals for given step sizes and high efficiency of the Markov chain; and (3) an automatic tuning scheme for adjusting the hyperparameters in the cyclical schedules, allowing adaptability across diverse datasets with minimal tuning. We prove the non-asymptotic convergence and inference guarantee for our method in general discrete distributions. Extensive experiments demonstrate the superiority of our method in sampling complex multimodal discrete distributions.

## 1 Introduction

Discrete variables are common in many machine learning problems, highlighting the crucial need for efficient discrete samplers. Recent advances [Grathwohl et al., 2021, Zhang et al., 2022b, Sun et al., 2021, 2023b,a, Xiang et al., 2023] have leveraged gradient information in discrete distributions to improve proposal distributions, significantly boosting their efficiency. These advancements have set new benchmarks in discrete sampling tasks across graphical models, energy-based models, and combinatorial optimization [Goshvadi et al., 2023].

However, one major limitation of gradient-based methods is their susceptibility to becoming trapped in local modes [Ruder, 2016, Ziyin et al., 2021], which significantly reduces the accuracy and efficiency of sampling results. In continuous spaces, several strategies such as cyclical step sizes [Zhang et al., 2020], parallel tempering [Swendsen and Wang, 1986, Deng et al., 2020a], and flat histograms [Berg and Neuhaus, 1991, Deng et al., 2020b], have been proposed to address this issue. When it comes to discrete distributions, which are inherently more multimodal due to their discontinuous nature, the problem becomes even more severe. Despite the pressing need, there is a lack of methodology for gradient-based discrete samplers to effectively explore multimodal distributions. Current methods

38th Conference on Neural Information Processing Systems (NeurIPS 2024).

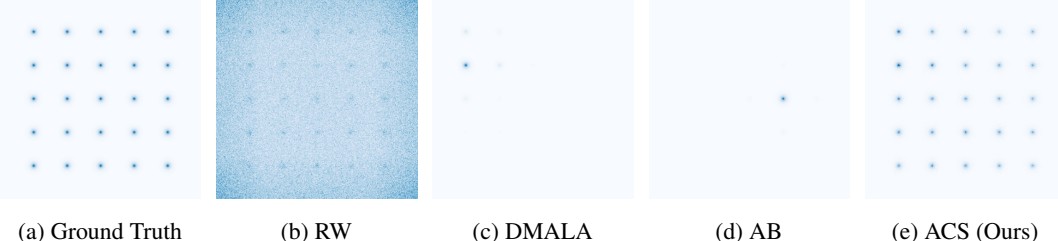

|  (a) Ground Truth | (b) RW | (c) DMALA | (d) AB | (e) ACS (Ours) |

Figure 1: Sampling on a 2d distribution with multiple modes. (a): ground truth. (b): results from a random walk sampler. (c): results from DMALA [Zhang et al., 2022b] with a manually tuned step size. (d): results from AB [Sun et al., 2023a]. (e): results from our method ACS. While the random walk sampler can find all modes, its characterization is noisy and lacks details for each mode. Gradient-based samplers (b) and (c) effectively characterize a specific mode but are easily trapped in some local modes. Our method (d) can find all modes efficiently and characterize each mode accurately.

often fall far short in traversing the complex landscapes of multimodal distributions, as illustrated in Figure 1.

In this paper, we propose *automatic cyclical scheduling* for gradient-based discrete sampling to efficiently and accurately sample from multimodal distributions. To balance between uncovering new modes and characterizing the current mode, we parameterize a family of gradient-based proposals that span a spectrum from local to global proposals. The parameterized proposal dynamically adjusts according to cyclical schedules of both step size and the balancing parameter, smoothly transitioning from global exploratory moves to more localized moves within each cycle. These cyclical schedules are automatically tuned by a specially designed algorithm, which identifies optimal step sizes and balancing parameters for discrete distributions. Our contributions are summarized as follows:

- We present the first gradient-based discrete sampling method that targets multimodal distributions. Our method incorporates cyclical schedules for both step size and balancing parameter to facilitate the exploration and exploitation in discrete distributions.

- We propose an automatic tuning algorithm to configure the cyclical schedule, enabling effortless and customized adjustments across various datasets without much manual intervention.

- We offer non-asymptotic convergence and inference guarantees for our method in general discrete distributions. To our knowledge, this is the first non-asymptotic convergence bound of gradient-based discrete sampling to the target distribution with inference guarantees, which could be of independent interest.

- We demonstrate the superiority of our method for both sampling and learning tasks including restricted Boltzmann machines, deep energy-based models, and large language models.

## 2   Related Work

**Gradient-based Discrete Sampling**   Zanella [2017] introduced a family of locally informed proposals, laying the foundation for recent developments in efficient discrete sampling. Building upon this, Grathwohl et al. [2021] further incorporates gradient approximation, significantly reducing computational costs. Following these pioneering efforts, numerous studies have proposed various gradient-based discrete sampling techniques [Rhodes and Gutmann, 2022, Sun et al., 2021, 2022, 2023b, Xiang et al., 2023]. Zhang et al. [2022b] develops a discrete Langevin proposal, translating the powerful Langevin algorithm to discrete spaces. Sansone [2022] introduces a self-balancing method to optimize the balancing functions in locally balanced proposals. While our work also utilizes an adaptive phase, it differs in that our parameterization extends beyond the local regime, and our proposal parameterization is considerably simpler.

Perhaps the most closely related study is the any-scale balanced sampler [Sun et al., 2023a]. This method uses a non-local balancing proposal and adaptively tunes it. Our work, however, differs in several key aspects: (1) We focus on combining both local and non-local proposals to effectively characterize multimodal discrete distributions, as opposed to focusing on a single optimal proposal. (2) Our automatic tuning algorithm adjusts the step size and balancing parameter by considering

the special discrete structures and targets a specific Metropolis-Hastings acceptance rate, rather than maximizing the average coordinates changed per step. (3) Our method can be applied to learning energy-based models (EBM) and sampling large language models, whereas their approach cannot.

**Sampling on Multimodal Distributions** There exist several sampling methods targeting discrete multimodal distributions, such as simulated tempering [Marinari and Parisi, 1992], the Swendsen-Wang algorithm [Swendsen and Wang, 1987], and the Wolff algorithm [Wolff, 1989]. However, these methods usually use random walk or Gibbs sampling as their proposals. It is unclear how these methods can be adapted for gradient-based discrete sampling.

In continuous spaces, various gradient-based methods have been developed specifically for multimodal distributions [Zhang et al., 2020, Deng et al., 2020a,b]. Our method distinguishes from the cyclical step size in Zhang et al. [2020] by incorporating an additional cyclical balancing parameter schedule and an automatic tuning scheme, which are crucial for efficient exploration in discrete distributions. Furthermore, our theoretical analysis of convergence is different from that in Zhang et al. [2020] which relies on continuous stochastic processes.

## 3 Preliminaries

### 3.1 Problem Definition

We consider the task of sampling from some target distribution defined over a discrete space

$$\pi(\theta) = \frac{1}{Z} \exp(U(\theta)), \quad \theta \in \Theta.$$

Here, $\theta$ is a $d$ dimensional discrete variable in domain $\Theta$, $U$ is the energy function, and $Z$ is the normalizing constant. We make the following assumptions of the domain and the energy function, following the literature of gradient-based discrete sampling [Grathwohl et al., 2021, Sun et al., 2021, Zhang et al., 2022b]: (1) The domain is coordinatewisely factorized, $\Theta = \Pi_{i=1}^{d}\Theta_i$. (2) The energy function $U$ can be extended to a differentiable function in $\mathbb{R}^d$.

### 3.2 Locally Balanced Proposals

Zanella [2017] introduces a family of informed proposals, which is defined below:

$$Q_{g,\alpha}(\theta'|\theta) = \frac{g\left(\frac{\pi(\theta')}{\pi(\theta)}\right) K_\alpha(\theta' - \theta)}{Z_{g,\alpha}(\theta)} \tag{1}$$

Here, $K_\alpha$ is a kernel that determines the scale of the proposal where $\alpha$ plays a similar role as the step size. $g(t)$ is a balancing function that determines how to incorporate the information about $\pi$. If $g(t) = tg(\frac{1}{t})$, the proposal becomes a locally balanced proposal, which is asymptotically optimal in the local regime, that is, when the step size $\alpha \to 0$.

## 4 Automatic Cyclical Sampler

We aim to develop a sampler capable of escaping local modes in general multimodal discrete distributions, including those that appear in deep energy-based models and large language models. First, we motivate using the cyclical schedule by demonstrating the issue of gradient-based samplers getting stuck in local modes on a toy dataset. We then present our sampler's parameterization of the step size and balancing function. Next, we introduce a cyclical schedule for the proposal distribution that enables effective exploration and characterization of discrete multimodal distributions. Finally, we develop an automatic tuning method that simplifies the process of identifying hyperparameters in cyclical schedules.

### 4.1 Motivating Example: A Synthetic Multimodal Discrete Distribution

To demonstrate the crucial issue of local modes trapping gradient-based samplers, we construct a 2-dimensional dataset consisting of integers. We define $\Theta = \{0, 1, \cdots N\}^2$, where $N$ is the maximum

value for each coordinate. Given a set of modes $\{\mu_1, \mu_2, \ldots \mu_l\}$, we define the energy as follows:

$$U(\theta) = \log\left(\sum_{i=1}^{l} \exp\left(\frac{||\theta - \mu_i||^2}{2\sigma}\right)\right). \tag{2}$$

This distribution enables easy comparison between different methods in terms of their ability to both explore and exploit the target distribution. We demonstrate the results of various samplers in Figure 1. More experimental details can be found in Appendix D.1.

A visual comparison reveals that while gradient-based samplers (DMALA [Zhang et al., 2022b] and AB [Sun et al., 2023a]) are very effective at characterizing a given mode, they tend to get trapped in some small neighborhood, preventing a proper characterization of the distribution as a whole.

We can understand this behavior of gradient-based samplers by comparing them to a random walk sampler (RW), which is able to explore all the modes but unable to fully characterize the detail of each one. While the RW sampler proposes movements uniformly over the sample space, gradient-based samplers propose movement based on the geometry of the distribution as captured by the gradient. Because the proposed movements are in the direction of increasing density, these proposals are able to characterize a given mode in detail. At the same time, these proposals hinder escape to more distant modes as the gradient points away from their direction. For this reason, we observe that local modes are able to "trap" gradient-based samplers.

## 4.2 Parameterized Proposal Distribution

To derive an automatic schedule for the proposal, we need to parameterize the proposal first. We define $K_\alpha$ and $g(t)$ in the informed proposal [Zanella, 2017] as follows:

$$K_\alpha(\theta' - \theta) = \frac{\exp\frac{-||\theta' - \theta||^2}{2\alpha}}{Z}, \quad \alpha \in (0, \infty); \qquad g(t) = t^\beta, \quad \beta \in [0.5, 1) \tag{3}$$

where $\beta$ is called a balancing parameter. $\alpha \to 0, \beta = 0.5$ correspond to a locally-balanced proposal and $\alpha \to \infty, \beta = 1$ correspond to a globally-balanced proposal. Values in between result in interpolations between locally-balanced and globally-balanced proposals. Note that $\beta \in (0, 1)$ in Sun et al. [2023a] while our range is narrower.

We substitute these definitions into Equation (1) and apply the first-order Taylor expansion:

$$Q_{\alpha,\beta}(\theta'|\theta) \propto \exp\left(\beta(U(\theta') - U(\theta)) - \frac{||\theta' - \theta||^2}{2\alpha}\right) \approx \exp\left(\beta(\nabla_\theta U(\theta)(\theta' - \theta)) - \frac{||\theta' - \theta||^2}{2\alpha}\right). \tag{4}$$

As in Zhang et al. [2022b], we use the assumption of coordinate-wise factorization to obtain the following coordinate-wise proposal function:

$$Q_{\alpha,\beta}^i(\theta'_i|\theta) = \text{Cat}\left(\text{Softmax}\left(\beta\nabla U(\theta)_i(\theta'_i - \theta_i) - \frac{(\theta'_i - \theta_i)^2}{2\alpha}\right)\right). \tag{5}$$

In order to make the resulting Markov chain reversible, we apply the Metropolis-Hastings correction, where we accept the proposed step with the following probability:

$$A(\theta'|\theta, \alpha, \beta) = \min\left(1, \exp(U(\theta') - U(\theta))\frac{Q_{\alpha,\beta}(\theta|\theta')}{Q_{\alpha,\beta}(\theta'|\theta)}\right). \tag{6}$$

In summary, we parameterize our proposal as in Equation (5) which includes a spectrum of local and global proposals. Our proposal is determined by two hyperparameters, the step size $\alpha$ and the balancing parameter $\beta$.

## 4.3 Cyclical Hyperparameter Schedules

**Cyclical Step Size Schedule**  In order to effectively explore the whole target distribution while retaining the ability to exploit local modes, we adopt the cyclical step size schedule from Zhang et al. [2020]. The definition of step size $\alpha$ for iteration $k$ is as follows:

$$\alpha_k = \max\left(\alpha_{\max} \cdot \cos\left(\frac{\pi\text{mod}(k, s)}{s}\right) + 1, \alpha_{\min}\right), \tag{7}$$

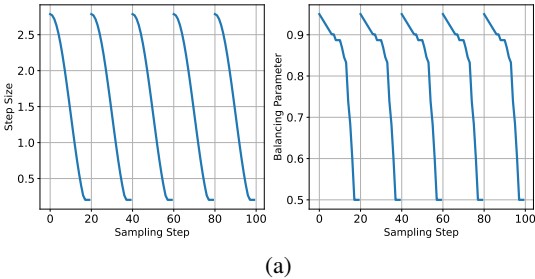
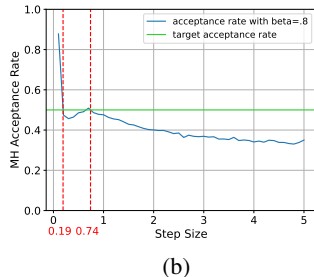

(a)                                                              (b)

Figure 2: (a) $\alpha$-schedule along with the corresponding $\beta$ schedule. The initial large steps enable the sampler to explore different regions of the distribution, while the smaller steps enable good characterization of each region. The balancing parameter $\beta$ varies with the step size to enable high acceptance rates for all step sizes. (b) Acceptance rate v.s. step size on EBM sampling on MNIST shows a non-monotonic relationship.

---

**Algorithm 1** Cyclical Sampling Algorithm

---

**Require:** step size schedule $\{\alpha_k\}_{k=1}^s$, balancing parameter schedule $\{\beta_k\}_{k=1}^s$, cycle number $n$, steps per cycle $s$
 1: samples $\leftarrow$ [ ]
 2: **for** cycle $c$ in range $n$ **do**
 3:     **for** step $k$ in range $s$ **do**
 4:         $\theta \leftarrow$ samples[-1]
 5:         **for** coordinate $i$ in range $d$ **do**
 6:             construct $Q_{\alpha_k,\beta_k}^i(\cdot|\theta)$ as in (5)
 7:             sample $\theta_i' \sim Q_{\alpha_k,\beta_k}^i(\cdot|\theta)$
 8:         **end for**
 9:         samples $\leftarrow \theta'$ with probability (6)
10:     **end for**
11: **end for**
12: **return** samples

---

where $\alpha_{\max}$ is the initial step size, $\alpha_{\min}$ is the minimum step size, and $s$ is the number of sampling steps per cycle. Differing from the cyclical schedule in Zhang et al. [2020], we additionally add $\alpha_{\min}$ to make sure that even the smallest step size remains effective in discrete spaces.

**Cyclical Balancing Schedule**   Using large step sizes in (7) can easily result in very low acceptance rates, removing any benefit of exploration. To address this issue, we introduce a balancing parameter schedule, which enables reasonable acceptance rates for large step sizes. As discussed in Zanella [2017], Sun et al. [2023a], the balancing parameter should vary with different step sizes to achieve a "balanced" proposal. A balanced proposal ensures that the Markov chain is reversible with respect to a certain distribution, which will converge weakly to the target distribution. For example, when the step size $\alpha \to 0$, the optimal balancing parameter is $\beta = 0.5$, whereas for $\alpha \to \infty$, the ideal balancing parameter becomes $\beta = 1$.

Thus for a schedule of step sizes, each $\alpha_i$ requires a different $\beta_i \in [.5, 1)$, with larger step sizes having $\beta_i$ closer to 1 and smaller step sizes having $\beta_i$ closer to 0.5. Using the Metropolis-Hastings acceptance rate to characterize the quality of a given $\alpha, \beta$ pair, we define the value of $\beta_i$ as follows:

$$\beta_i = \operatorname{argmax}_{\beta \in [.5, \beta_{i-1}]} \left( \mathbb{E}_{\theta \sim \pi, \theta' \sim Q_{\alpha,\beta}} \left[ \{ A(\theta'|\theta, \alpha_i, \beta) \} \right] \right) \tag{8}$$

Intuitively, this definition means that the best $\beta_i$ for a given step size $\alpha_i$ maximizes the average acceptance rate for the proposal function $Q_{\alpha,\beta}$. It also conveys that larger step sizes will have larger balancing parameters.

We include a visualization of the resulting schedules in Figure 2a and outline our algorithm using the $\alpha, \beta$ schedules in Algorithm 1. Note that it incurs no extra overhead compared to previous gradient-based discrete sampling methods as it only adjusts hyperparameters $\alpha$ and $\beta$. By using a combination of large and small $\alpha$ and $\beta$, we enable the sampler to explore the distribution fully without sacrificing the ability to characterize each mode. This is demonstrated in Figure 1e.

---

**Algorithm 2** Automatic Schedule Tuning Algorithm

---

**Require:** $\beta_{\min} = .5$, $\beta_{\max}$, target acceptance rate $\rho^*$, initial state $\theta_{\text{init}}$, steps per cycle $s$, initial largest step size $\alpha_{\text{ceil}} = 60$, initial smallest step size $\alpha_{\text{floor}} = .05$
1:  $\theta \leftarrow \text{InitBurnin}(\alpha_{\text{ceil}}, \beta_{\max}, \theta_{\text{init}})$
2:  $\alpha_{\min} \leftarrow \text{EstAlpha}(\alpha_{\text{floor}}, \beta_{\min}, \theta, \rho^*, \text{MAX=False})$
3:  $\alpha_{\max} \leftarrow \text{EstAlpha}(\alpha_{\text{ceil}}, \beta_{\max}, \theta, \rho^*, \text{MAX=True})$
4:  Construct $\alpha$-sched of length $s$ using (7)
5:  $\beta$-sched $\leftarrow \text{EstBalSched}(\alpha\text{-sched}, \beta_{\max}, \beta_{\min}, \theta)$
6:  **return** $\alpha$-sched, $\beta$-sched

---

### 4.4 Automatic Schedule Tuning

For schedules in Equations (7) and (8), we have parameters $\alpha_{\max}$, $\alpha_{\max}$, and $\{\beta_1, \beta_2 \ldots \beta_s\}$ to be decided. In this section, we will introduce an automatic tuning algorithm to easily find suitable values.

**Main Idea**   Our automatic tuning algorithm depends on the initial balancing parameter $\beta_{\max}$, the final balancing parameter $\beta_{\min}$, a target acceptance rate $\rho^*$, and the number of steps per cycle $s$. These values are relatively easy to select, as detailed in Appendix A. Below, we assume they are already determined. The tuning algorithm first estimates the optimal choices for $\alpha_{\max}$ and $\alpha_{\min}$ based on $\rho^*$, which can then be used to construct the full step-size schedule using (7). We then construct the balancing parameter schedule using (8). The method is summarized in Algorithm 2 with details regarding subroutines in Appendix A. Our automatic tuning introduces minimal overhead relative to the more expensive sampling process. For example, in Section 6, we use 500 steps as the budget for Algorithm 2 where the total number of sampling steps is at least 5000. We further demonstrate that our algorithm is relatively robust to hyperparameters in Appendix A.1.

In short, our tuning algorithm adopts an alternative optimization strategy, leveraging existing knowledge about hyperparameter values (e.g. $\beta_{\min}$ and $\beta_{\max}$ should be around 0.5 and 1 respectively). While estimating the best pair $\alpha, \beta$ is challenging due to their interdependence, it is much easier to fix one and optimize the other [Sun et al., 2023a].

**Estimating $\alpha_{\max}, \alpha_{\min}$**   For a given $\beta_{\max}, \beta_{\min}$, our goal is to find step sizes $\alpha_{\max}, \alpha_{\min}$ that enable an acceptance rate close to $\rho^*$. We can formally state this goal as follows:

$$J(\alpha, \beta) = \mathbb{E}_{\theta \sim \pi}\left[\mathbb{E}_{\theta' \sim Q_{\alpha,\beta}(\cdot|\theta)}\left|\rho^* - A(\theta'|\theta, \alpha, \beta)\right|\right]. \tag{9}$$

Given $\beta_{\max}, \beta_{\min}$, we construct the following objectives to pick the corresponding $\alpha_{\max}, \alpha_{\min}$:

$$\begin{aligned}\alpha_{\max} &= \max\{\alpha \text{ s.t } J(\alpha, \beta_{\max}) \approx 0\}\\ \alpha_{\min} &= \min\{\alpha \text{ s.t } J(\alpha, \beta_{\min}) \approx 0\}.\end{aligned} \tag{10}$$

By defining the initial and final step sizes in this manner, we ensure that our cyclical schedule includes a wide range of hyperparameter pairs with different trade-offs in exploration and exploitation.

To solve (10), we estimate $\alpha_{\max}$ by starting with a large step size and gradually decreasing it to find the step size that yields $\rho^*$. Unlike existing works that start with small step sizes, we observed that multiple $\alpha$ values can yield the same acceptance rate for a given $\beta$, as shown in Figure 2b. Therefore, we start with an upper limit $\alpha_{\text{ceil}}$ and reduce the step size to avoid missing any larger $\alpha$ values that meet our criteria. Detailed implementation is provided in Algorithm 4 in the Appendix. $\alpha_{\min}$ can be obtained similarly.

**Estimating Balancing Schedule**   After setting the start and end pairs for the $\alpha$ and $\beta$ schedules, we now define intermediate $\beta$ values. As the entire step size schedule is fixed by (7), the problem is to determine the best balancing parameter for each step size. A simple strategy is to test different $\beta$ spaced out evenly throughout the interval $[.5, \beta_{i-1}]$ and select the best value in terms of acceptance rate. This approach leverages the observation that smaller step sizes tend to have smaller optimal balancing parameters. Detailed implementation is given in Algorithm 5 in Appendix.

# 5 Theoretical Analysis

In this section, we present a convergence rate analysis for Algorithm 1. For general step size and balancing parameter schedules, i.e., at each cycle, the algorithm will go through $s$ steps in which it will use step sizes $\alpha_1, \alpha_2, \cdots, \alpha_s$ and balancing parameters $\beta_1, \beta_2, \cdots, \beta_s$. Note that for each pair $(\alpha_i, \beta_i)$, we have a Markov transition operator which we label $P_i$ for $i = 1, 2, \cdots, s$. The Markov operator for a single cycle is given by $\hat{P} = P_1 P_2 \cdots P_s$. We have the following two assumptions:

**Assumption 5.1.** The function $U(\cdot) \in C^2(\mathbb{R}^d)$ has $M$-Lipschitz gradient. That is

$$\|\nabla U(\theta) - \nabla U(\theta')\| \leq M \|\theta - \theta'\|.$$

Note that it implicitly assumes that the set in domain $\Theta$ is finite. We define $conv(\Theta)$ as the convex hull of the set $\Theta$.

**Assumption 5.2.** For each $\theta \in \mathbb{R}^d$, there exists an open ball containing $\theta$ of some radius $r_\theta$, denoted by $B(\theta, r_\theta)$, such that the function $U(\cdot)$ is $m_\theta$-strongly concave in $B(\theta, r_\theta)$ for some $m_\theta > 0$.

Assumptions 5.1 and 5.2 are standard in optimization and sampling literature [Bottou et al., 2018, Dalalyan, 2017]. Under Assumption 5.2, $U(\cdot)$ is $m$-strongly concave on $conv(\Theta)$, following Lemma C.3 in Appendix.

We define $diam(\Theta) = \sup_{\theta, \theta' \in \Theta} \|\theta - \theta'\|$ and $\epsilon_{\alpha_i, \beta_i}$ to be

$$\exp\left\{ -\left( \frac{1}{2\alpha_i} + \beta_i M - \frac{\beta_i m}{2} \right) diam(\Theta)^2 - \|\nabla U(a)\| diam(\Theta) \right\}.$$

The Markov kernel corresponding to each $P_i$ in each step of the cycle in Algorithm 1 is

$$p_i(\theta'|\theta) = A(\theta'|\theta, \alpha_i, \beta_i) Q_{\alpha_i, \beta_i}(\theta'|\theta) + (1 - L(\theta)) \delta_\theta(\theta') \tag{11}$$

where

$$L(\theta) = \sum_{\theta' \in \Theta} \left( \frac{\pi(\theta') Q_{\alpha_i, \beta_i}(\theta|\theta')}{\pi(\theta) Q_{\alpha_i, \beta_i}(\theta'|\theta)} \wedge 1 \right) Q_{\alpha_i, \beta_i}(\theta'|\theta)$$

is the total rejection probability from $\theta$. Finally, recall that the total variation distance between two probability measures $\mu$ and $\nu$, defined on some space $\Theta \subset \mathbb{R}^d$ is

$$\|\mu - \nu\|_{TV} = \sup_{A \in \mathcal{B}(\Theta)} |\mu(A) - \nu(A)|$$

where $\mathcal{B}(\Theta)$ is the set of all measurable sets in $\Theta$.

**Constant Step Size and Balancing Parameter**  To analyze Algorithm 1 with step size and balancing parameter schedules, we first solve a simpler problem where the step size and balancing parameter are fixed and then extend the analysis to the setting of Algorithm 1.

Our main method of proof is to establish uniform ergodicity of the Markov chain $P$, for a single $\alpha, \beta$, by establishing a uniform minorization for $P$. We denote the transition kernel for this Markov chain $P$ as $p(\cdot \mid \cdot)$, which is given in (11) with $\alpha_i, \beta_i$ replaced by a fixed $\alpha, \beta$.

**Lemma 5.3.** *Let Assumptions 5.1-5.2 with $\alpha < \frac{1}{\beta M}$ hold. Then for the Markov chain $P$ we have, for any $\theta, \theta' \in \Theta$,*

$$p(\theta \mid \theta') \geq \epsilon_{\beta, \alpha} \frac{\exp\{\beta U(\theta')\}}{\sum_{\theta' \in \Theta} \exp\{\beta U(\theta')\}},$$

*where*

$$\epsilon_{\beta, \alpha} = \exp\left\{ -\left( \frac{1}{2\alpha} + \beta M - \frac{\beta m}{2} \right) diam(\Theta)^2 \right.$$
$$\left. - \|\nabla U(a)\| diam(\Theta) \right\}$$

*with $a \in \arg\min_{\theta \in \Theta} \|\nabla U(\theta)\|$.*

*Proof.* The proof is provided in Appendix C.1. $\square$

**Theorem 5.4.** *Let Assumptions 5.1-5.2 hold with $\alpha < 1/\beta M$. Then for the Markov chain $P$, the following hold:*
*i. $P$ is uniformly ergodic with*

$$\|P^n - \pi\|_{TV} \le (1 - \epsilon_{\beta,\alpha})^n.$$

*ii. For any real-valued function $f$ and samples $X_1, X_2, X_3, \cdots, X_n$ from $P$, one has*

$$\sqrt{n}\left(\frac{1}{n}\sum_{i=1}^{n} f(X_i) - \sum_{\theta \in \Theta} f(\theta)\pi(\theta)\right) \xrightarrow{d} N(0, \tilde{\sigma}_*^2)$$

*for some $\tilde{\sigma}_* > 0$ as $n \to \infty$.*

*Proof.* The proof directly follows from our Lemma 5.3 and Jones [2004][Corollary 5]. □

Note that as $\alpha \to 0$, we have $\epsilon_{\beta,\alpha} \to 1$ which implies that small step sizes result in low convergence rates. This is intuitive as the algorithm could not explore much in this case. Furthermore, our results suggest that large $\beta$ restricts $\alpha$ to small values. Given that large $\beta$ generally requires large $\alpha$, our findings imply an upper bound for the step size.

**Adaptive Step Size and Balancing Parameter**    Now we tackle the case of varying step sizes and balancing parameters. Each cycle has $s$ steps with step sizes $\alpha_1, \alpha_2, \cdots, \alpha_s$ and balancing parameters $\beta_1, \beta_2, \cdots, \beta_s$. Note that this case is more challenging as at each step the transition operator changes and the Markov chain is no longer homogeneous. However, the marginal chain for each cycle is indeed homogeneous and can be analyzed. We present our results in this setting as follows:

**Theorem 5.5.** *Let Assumptions 5.1 and 5.2 hold with $\alpha_i < 1/\beta_i M$, $i = 1, 2, \cdots s$. Then for the Markov chain $\hat{P}$, the following hold*
*i. $\hat{P}$ is uniformly ergodic with*

$$\left\|\hat{P}^n - \pi\right\|_{TV} \le (1 - \epsilon_{\beta_s,\alpha_s})^n.$$

*ii. For any real-valued function $f$ and samples $X_1, X_2, X_3, \cdots, X_n$ from $\hat{P}$, one has*

$$\sqrt{n}\left(\frac{1}{n}\sum_{i=1}^{n} f(X_i) - \sum_{\theta \in \Theta} f(\theta)\pi(\theta)\right) \xrightarrow{d} N(0, \tilde{\sigma}_*^2)$$

*for some $\tilde{\sigma}_* > 0$ as $n \to \infty$, where,*

$$\epsilon_{\beta_s,\alpha_s} = \exp\left\{-\left(\frac{1}{2\alpha_s} + \beta_s M - \frac{\beta_s m}{2}\right) diam(\Theta)^2\right\}$$
$$\cdot \exp\left\{-\|\nabla U(a)\| diam(\Theta)\right\}$$

*with $a \in \arg\min_{\theta \in \Theta}\|\nabla U(\theta)\|$.*

*Proof.* The proof follows from our Lemma 5.3, Proposition C.1 and Jones [2004][Corollary 5]. □

Both Theorems 5.4 and 5.5 hold uniformly over all functions in the class of functions with at least a local minima in $\Theta$. The Central Limit Theorem results in Theorems 5.4 and 5.5 imply that we may perform inference on the target distribution $\pi(\cdot)$ even though the asymptotic variances are unknown, as we may perform batch-means to estimate these variances Vats et al. [2019].

In summary, we have established a geometric convergence rate to the target distribution for our sampler. Previous research has only established asymptotic convergence [Zhang et al., 2022b] or relative convergence rate bounds [Grathwohl et al., 2021] for gradient-based discrete samplers. To the best of our knowledge, our results present the first non-asymptotic convergence bounds that explicitly quantify the distance between the estimated and target distributions. Further, our convergence bound also shows that discrete spaces play a fundamental part in the ergodic nature of these algorithms.

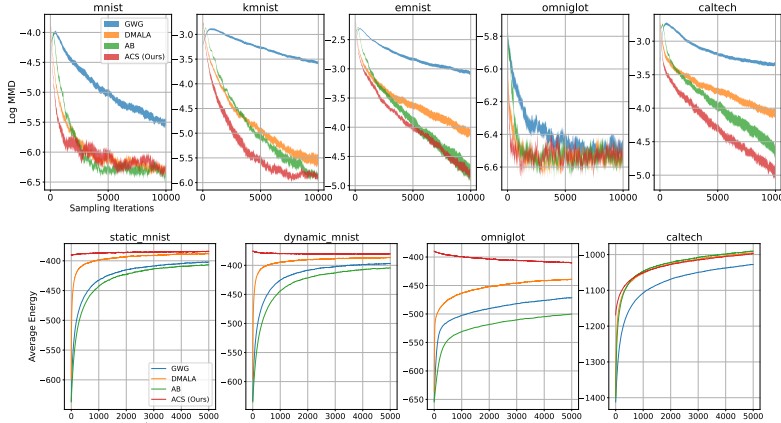

Figure 3: Sampling performance of various methods. Top row demonstrates convergence to ground truth on RBMs, bottom row demonstrates convergence speed on deep EBMs. We report the average performance across 11 random seeds within 1 standard error for the top row, and we show the average performance for the bottom row, as the error area is not visibly clear. For both distribution types, ACS demonstrates competitive performance with all baselines.

# 6 Experiments

We call our method that combines Algorithm 1 and 2 *Automatic Cyclical Sampler* (ACS). For RBM and EBM sampling tasks, we compare our method to Gibbs-with-Gradient (GWG) [Grathwohl et al., 2021], Any-scale sampler (AB) [Sun et al., 2023a], and Discrete Metropolis Adjusted Langevin Algorithm (DMALA) [Zhang et al., 2022b], which are popular and recent gradient-based discrete samplers. For learning tasks, we omit AB sampler as it is not originally applied to the model learning tasks. More experimental details are in Appendix D. We released our code at the following link: `https://github.com/patrickpynadath1/automatic_cyclical_sampling`.

## 6.1 Sampling Tasks

We evaluate our sampling method on both Restricted Boltzmann Machines (RBMs) and deep convolutional Energy-Based Models (EBMs). For RBMs, we measure accuracy by comparing the Maximum Mean Divergence (MMD) between samples generated by our method and Block Gibbs, which can be considered the ground truth. We sample on EBMs to demonstrate our method's scalability to more complex distributions. Experimental details are provided in Appendices D.2 and D.3 for RBM and EBM sampling, respectively.

**Results** In Figure 3, our proposed ACS method performs competitively for both RBMs and EBMs across all datasets. For RBM sampling, ACS is able to converge to the ground truth quicker than other methods due to the ability to capture the multi-modal nature of the target distribution. We see that this performance generalizes to more complex distributions as represented by deep EBMs.

## 6.2 Learning RBMs and EBMs

One common application of MCMC techniques is learning energy-based models (EBMs), where a neural network parameterized by $\phi$ represents an energy function $E_\phi$. These models are typically trained using Persistent Contrastive Divergence (PCD) and evaluated with Annealed Importance Sampling (AIS). Details on ACS for EBM learning are in Appendix B. We test our algorithm on

Table 1: Deep Convolution EBM Log likelihood scores on test data as estimated by AIS. GWG results are taken from [Grathwohl et al., 2021]. ACS is able to achieve better results than the baselines.

|  | GWG* | DMALA | ACS |
|---|---|---|---|
| Static MNIST | −80.01 | −80.031 ± 0.038 | **−79.905 ± 0.057** |
| Dynamic MNIST | −80.51 | −80.120 ± 0.036 | −79.634 ± 0.024 |
| Omniglot | −94.72 | −99.243 ± 2.101 | **−91.487 ± 0.128** |
| Caltech | −96.20 | −98.001 ± 0.371 | **−89.262 ± 0.290** |

learning deep convolutional EBMs, with experimental details in Appendix D.5. We include additional experimentation with learning RBMs in Appendix D.4.

**Results** Table 1 demonstrates that ACS is capable of learning better quality EBMs given the same computational budget as DMALA. Furthermore, ACS learns better quality models with *less* computational budget than GWG.

## 6.3   Text Infilling

One challenging application of discrete MCMC methods is text-infilling, where the goal is to complete a sentence with some missing words. Given a dataset of sentences, we randomly mask our 50% of the words and fill them in using the distribution given by a pretrained RoBERTa model. We include experiment details in Appendix D.6.

**Results**   Table 2 demonstrates that ACS is capable of generating more diverse sentences, as ACS has a lower self-BLEU and higher percentage of unique n-grams. While the perplexity results seem to imply that ACS generates lower quality than DMALA, we note that the ACS generations are more likely to be predicted as linguistically acceptable as shown by the CoLA scores. We discuss the results more extensively in Appendix D.6.

| Dataset | Method | Perplexity ($\downarrow$) | CoLA ($\uparrow$) | Self-Bleu ($\downarrow$) | Unique n-gram ($\uparrow$) | |
|---|---|---|---|---|---|---|
| | | | | | n=2 | n=3 |
| Grimm | DMALA | $\mathbf{280.82 \pm 27.26}$ | $50.46 \pm 1.25$ | $41.83 \pm 6.85$ | 48.55 | 70.56 |
| | ACS | $369.44 \pm 30.85$ | $\mathbf{53.42 \pm 1.26}$ | $\mathbf{36.70 \pm 6.42}$ | **53.91** | **74.70** |
| SST2 | DMALA | $\mathbf{256.66 \pm 10.53}$ | $42.62 \pm 1.14$ | $37.47 \pm .79$ | 57.68 | 75.21 |
| | ACS | $307.05 \pm 14.84$ | $\mathbf{47.12 \pm 1.20}$ | $\mathbf{32.42 \pm .75}$ | **62.54** | **78.87** |

Table 2: Empirical evaluation of the generated sentences. ACS outperforms DMALA for all metrics related to diversity.

## 7   Conclusion and Limitations

In this work, we propose Automatic Cyclical Sampler (ACS) to more effectively characterize multimodal distributions in discrete spaces. First, we demonstrate that gradient-based samplers are prone to getting trapped in local modes, preventing a full characterization of target distributions. To address this issue, we combine a cyclical step size schedule with a cyclical balancing parameter schedule along with an automatic tuning algorithm to configure these schedules. We also theoretically establish the non-asymptotic convergence bound of our method to the target distribution in addition to providing extensive experimental results.

While our proposed ACS method generates impressive results on a wide range of experiments, there are some limitations to our work that should be mentioned. Specifically, though we have proven a geometric convergence rate and the relationship between $\alpha$ and $\beta$ in our theoretical analysis, we require $U(\cdot)$ to be twice differentiable as well as locally strongly concave and the proof is not based on the specific tuning algorithm implemented. This is why we provide extensive experimentation to demonstrate that our algorithm is capable of picking good $\alpha, \beta$ schedules.

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

## A Details of Automatic Cyclical Sampler Algorithm

Here we include more details regarding the Automatic Cyclical Sampler algorithm. We discuss all the individual sub-routines that compose the algorithm shown in Algorithm 2. We also include an ablation study to demonstrate the robustness of our algorithm to various hyper-parameter configurations.

**InitialBurnin**   We find that in order to produce meaningful estimates for the objective in (9), it is necessary to burn in the MCMC sampling chain. This is due to the dependence of the acceptance rate on current sample $\theta$. If we use $\theta$ very low in density with respect to the target distribution, the acceptance rates estimated by the tuning algorithm will lose accuracy as the sampler converges to the target distribution. In order to avoid this issue, we run a quick burn-in stage with two distinct stages.

The first stage uses the gradient information to move the sampler away from the initialized point as quickly as possible. We use the parameterized proposal from Equation (4) with stepsize $\alpha_{\text{ceil}}, \beta_{\text{max}}$ without any Metropolis-Hastings correction as this enables very large movements from the initial sample.

For some datasets, this enables a very quick burn-in. This can be noticed in Figure 3 for Static/Dynamic MNIST and Omniglot. We hypothesize that this is due to the distribution having a relatively simple structure that enables the gradient to provide meaningful information for very large sampling steps. It is impossible to determine *a priori* whether a given distribution will have this property, so we include a following stage that uses a Metropolis-Hastings correction to increase the chance of arriving at a reasonable sample $\theta$.

For this stage, we construct a naive step size schedule and balancing constant schedule using the values of $\alpha_{\text{ceil}}, \alpha_{\text{floor}}, \beta_{\text{max}}, \beta_{\text{min}}$. We then run the parameterized sampler from Equation (4) with the Metropolis-Hastings correction. Our goal is to move the sampler to samples $\theta$ that are more likely in the target distribution. This will enable the acceptance rates computed during the tuning algorithm to be closer to the acceptance rates for the steady-state chain.

For all the sampling experiments, these two stages combined use 100 sampling steps.

**EstimateAlpha**   Here we discuss the algorithm used to calculate both $\alpha_{\text{max}}, \alpha_{\text{min}}$ as defined in Equation (10). When calculating $\alpha_{\text{max}}$, the goal is to pick the largest stepsize $\alpha_{\text{max}}$ that acheives the acceptance rate $\rho^*$ for a given $\beta_{\text{max}}$. When calculating $\alpha_{\text{min}}$, the goal is to determine the smallest step-size capable of acheiving the target acceptance rate. We put the full pseudo-code in Algorithm 4.

For calculating $\alpha_{\text{max}}$ and $\alpha_{\text{min}}$, the algorithm follows the general pattern of automatically shifting the range of potential $\alpha$ based on the best values calculated from the previous iteration. When calculating $\alpha_{\text{max}}$, the algorithm starts with an upper-bound initialized to $\alpha_{\text{bound}} = \alpha_{\text{ceil}}$ and iteratively decreases the range of proposed $\alpha$. For $\alpha_{\text{min}}$, the algorithm starts with a lower bound $\alpha_{\text{bound}} = \alpha_{\text{floor}}$ and iteratively increases the range. For both, the other bound is calculated by the following learning rule:

$$\alpha_{\text{prop}} = \alpha_{\text{bound}} \pm \zeta |\rho - \rho^*|.$$

Here, $\zeta$ is the learning rate that determines how much we can adjust the step size in one tuning step. We found $\zeta$ insensitive and set $\zeta = .5$ in all tasks. Additionally, $\rho$ is the best acceptance rate computed from the previous iteration of the algorithm. For the first step of the algorithm, we set $\rho = 0$.

The algorithm uses $\alpha_{\text{prop}}, \alpha_{\text{bound}}$ to determine the range of $\alpha$ to test. For calculating $\alpha_{\text{max}}$, the algorithm searches in the range of $[\alpha_{\text{prop}}, \alpha_{\text{bound}}]$. For calculating $\alpha_{\text{min}}$, the range is $[\alpha_{\text{bound}}, \alpha_{\text{prop}}]$.

Given the appropriate range of $\alpha$ and an initial $\theta$, we test $t$ potential $\alpha$ and calculate their respective acceptance rates using Equation (6). Once we have computed all the acceptance rates, we set $\alpha_{\text{bound}}$ to the value that resulted in the most optimal acceptance rate as determined by Equation (9), $\theta$ to the corresponding $\theta'$, and $\rho$ to the corresponding acceptance rate.

**Choice of $\beta_{\text{max}}, \beta_{\text{min}}, \rho^*, s$**   The automatic tuning algorithm depends on an initial choice of $\beta_{\text{max}}, , \beta_{\text{min}}, \rho^*, s$ that enable it to automatically configure an effective hyper-parameter schedule. Here we describe the general approach to picking these values.

For some target distributions, it is possible that the best possible acceptance rate with a very high $\beta_{\text{max}}$, such as $\beta_{\text{max}} = .95$, will not be close to the target acceptance rate $\rho^*$. In this case, the

EstimateAlphaMax algorithm will keep on decreasing the proposed $\alpha_{\max}$, which will result in a very small $\alpha_{\max}$. In order to avoid this behavior, we recommend starting with $\beta_{\max} = .95$, and decreasing it by .05 if the resulting $\alpha_{\max}$ is reasonable.

We always set $\beta_{\min} = 0.5$ which is the smallest value $\beta$ can take.

We determine the target $\rho^*$ by starting with a value of .5 and increasing it by .05 until desirable performance metrics are obtained. While this process is essentially the same as a grid search, we note that we only needed to apply this process in the specific case of training a deep EBM on the Caltech Silhouettes dataset. For all other tasks and datasets, the target acceptance rate of $\rho^* = .5$ was effective. We discuss the unique difficulty presented within the Caltech Silhouettes dataset in D.5.

To determine the steps per cycle $s$, we required a similar approach to determine the optimal value. In our experiments, we only look at two different values: either $s = 8$, or $s = 20$. Having a longer cycle length tends to enable more exploitation of the target distribution, whereas having a shorter cycle enables more exploration. While we do not have an algorithm for automatically configuring this value, we were able to achieve good results across all tasks and datasets by choosing either of these two values. For more details on the resulting hyper-parameters used for each experiment, see Appendix D.

## A.1  Hyper-parameter Sensitivity

Our method introduces the following hyperparameters: $\beta_{\max}$, $\beta_{\min}$, $\alpha_{\text{ceil}}$, $\alpha_{\text{floor}}$, learning rate for tuning $\gamma$, steps per cycle $s$, target acceptance rate $\rho^*$, and budget $B$. This may seem like many additional hyperparameters, but the majority of these are introduced due to the automatic tuning mechanism and are not changed across all tasks and datasets in the paper: $\gamma = .5$, $\alpha_{\text{floor}} = .05$, $\alpha_{\text{ceil}} = 5$, $\beta_{\min} = .5$, $B = 200$. Thus the only hyperparameters requiring tuning in practice are $\beta_{\max}$, $\rho^*$, and $s$. Note that the existing adaptive discrete sampler, any-scale sampler introduced in [Sun et al., 2023a], has a similar number of hyper-parameters: initial step size $\sigma$, initial balancing parameter $\alpha$, update rate $\gamma$, decay rate $\beta$, buffer size $N$, initial Hessian matrix $W$, and initial diagonal matrix $D$. Like our method, most of these hyperparameters are fixed across experiments.

We conduct an ablation study to evaluate the sensitivity of our tuning algorithm to these hyperparameters choices. We choose one hyperparameter at a time to ablate and keep the rest at default values of the hyperparameters at their default setting. We run the RBM sampling experiment over multiple datasets, each for 10 random seeds, and report the average results in Figure 4. We omit the standard error as that would harm the interpretability of the graph as many of the plots are quite close together.

We can summarize the main takeaways as follows:

1. The sensitivity of our algorithm to the hyperparameters depends on the dataset. For example, the sensitivity of our algorithm is low on MNIST, kMNIST, eMNIST, and Omniglot while the sensitivity is relatively high on Caltech.

2. The optimal hyperparameter values depend on the dataset. For example, high values of $s$ generally yield superior results, except for Caltech, where lower values excel. Similarly, low $\beta_{\max}$ values are usually less effective, though Caltech is an exception, showcasing decent outcomes. In general, the hyperparameter values we selected to generate the final results in the experiment section were the ones that generalized across the datasets.

3. For each ablation, the values tested demonstrate reasonable results when compared with the baselines. While not all hyperparameter values result in equally competitive performance, all of them outperform the Gibbs-With-Gradient sampler Grathwohl et al. [2021]. This demonstrates that our method performs well with a wide range of hyperparameters and can achieve even better performance with careful hyperparameter tuning.

In conclusion, we believe these results demonstrate that our algorithm is relatively robust to choice in hyperparameters.

---

**Algorithm 3** InitBurnin

---

**Require:** $\alpha_{\text{ceil}}, \alpha_{\text{floor}}, \beta_{\max}, \beta_{\min}$, steps per cycle $s$, steps to take without MH correction $l = 50$, steps to take with MH correction $l_{\text{MH}} = 50$, initial state $\theta$

1: **for** step $i$ in range($l$) **do**
2:     $\theta \sim Q_{\alpha_{\text{ceil}}, \beta_{\max}}(\cdot|\theta) \triangleright$ *Run burnin steps without MH correction*
3: **end for**
4: $\{\alpha_0, \alpha_1 \cdots \alpha_{s-1}\} \leftarrow$ values from Equation (7) using $\alpha_{\text{ceil}}, \alpha_{\text{floor}}$.
5: $\{\beta_0, \beta_1 \cdots \beta_{s-1}\} \leftarrow$ values from Equation (7) using $\beta_{\max}, \beta_{\min} \triangleright$ *We can use Equation (7) to get interpolations of $\beta$*
6: number of cycles $n = \text{floor}(\frac{l_{\text{MH}}}{s})$
7: Obtain $\theta$ by running Algorithm 1 using the calculated $\alpha, \beta$ schedule $\triangleright$ *Run burnin steps with MH correction*
8: **return** $\theta$

---

---

**Algorithm 4** EstimateAlpha

---

**Require:** $\alpha_{\text{bound}}$, BUDGET, initial state $\theta$, Balancing parameter $\beta$, target acceptance rate $\rho^*$, learning rate $\zeta$, number of proposals per step $t = 5$, flag MAX

1: $\rho_{\text{cur}} \leftarrow 0$
2: **while** iteration $i \leq$ BUDGET **do**
3:     **if** MAX **then**
4:        $\alpha_{\text{prop}} = \alpha(1 - \zeta|\rho^* - \rho_{\text{cur}}|) \triangleright$ *adaptively decrease the range of potential $\alpha$*
5:        proposed-params $\leftarrow$ LinSpace($\alpha_{\text{prop}}, \alpha_{\text{bound}}, t) \triangleright$ *we use $\alpha_{bound} = \alpha_{ceil}$ as the ceiling for proposed $\alpha$*
6:     **else**
7:        $\alpha_{\text{prop}} = \alpha(1 + \zeta|\rho^* - \rho_{\text{cur}}|) \triangleright$ *For AlphaMin, adaptively increase the range of potential $\alpha$*
8:        proposed-params $\leftarrow$ LinSpace($\alpha_{\text{bound}}, \alpha_{\text{prop}}, t) \triangleright$ *For AlphaMin, use $\alpha_{bound} = \alpha_{floor}$ as the floor for proposed $\alpha$*
9:     **end if**
10:     initialize bookkeeping to keep track of proposed states and acceptance rates
11:     **for** $\alpha \in$ proposed-params **do**
12:        $\theta' \sim Q_{\alpha_{\text{prop}}, \beta}(\cdot|\theta) \triangleright$ *Use proposed $\alpha$ to take sampling step*
13:        $\rho = A(\theta'|\theta, \alpha_{\text{prop}}, \beta) \triangleright$ *Compute acceptance rate for proposed $\alpha$*
14:        $i = i + 1$
15:     **end for**
16:     Set $\rho_{\text{cur}}$ to the acceptance rate closest to the target $a^*$
17:     Set $\alpha_{\text{bound}}$ to the corresponding $\alpha \triangleright$ *Update $\alpha_{bound}$ to shift the range of proposed $\alpha$ for the next step*
18:     set $\theta_{\text{cur}}$ to the corresponding $\theta$
19: **end while**
20: **if** MAX **then**
21:     return $\alpha_{\max} = \alpha_{\text{bound}}$
22: **else**
23:     return $\alpha_{\min} = \alpha_{\text{bound}}$
24: **end if**

---

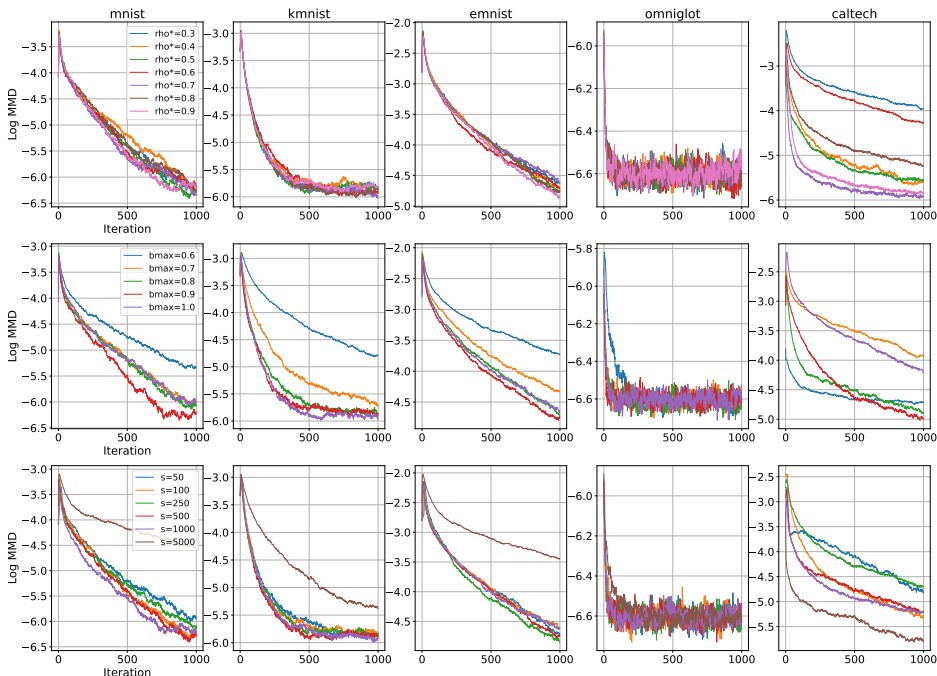

Figure 4: Average performance across multiple seeds for various hyper-parameter settings. We note that all configurations to exhibit convergence to the ground truth as indicated by the maximum mean discrepancy (log MMD), albeit with varying convergence speeds. In some cases, specific hyper-parameter configurations are able to achieve better performance than what we report in the RBM sampling experiment. Overall, we can observe that our algorithm is reasonably robust to various hyper-parameter configurations as it will still demonstrate convergent behavior towards the ground truth.

# B   ACS for EBM Learning

## B.1   Background

Energy Based Models (EBMs) are a class of generative models that learn some unnormalized distribution over a sample space. As discussed in Hinton [2002], these models can be trained via the following Maximum Likelihood objective:

$$\mathcal{L}(\phi) = \mathbb{E}_{x \sim p_{\text{data}}} \left[ -\log p_\phi(x) \right] \tag{12}$$

The gradient updates for this loss function are known to be as follows:

$$\nabla_\phi L(\phi) = \mathbb{E}_{x \sim p_{\text{data}}} \left[ \nabla_\phi E_\phi(x) \right] - \mathbb{E}_{x \sim p_\phi} \left[ \nabla_\phi E_\phi(x) \right] \tag{13}$$

While the expectation on the left is straight forward to calculate given a dataset, calculating the right expectation is not as clear. Here we will mention the two methods that are relevant towards our experiments with EBMs.

**Contrastive Divergence (CD)**    In order to estimate the second term, we initialize some sampler using the $x$ in the first term and run it for a set number of sampling steps. For a more detailed description, refer to Hinton [2002].

**Persistent Contrastive Divergence (PCD)**    The expectation on the right can be calculated using samples from a persistent Markov Chain that approximates the true distribution Tieleman [2008].

---

**Algorithm 5** EstimateBalSched

---

**Require:** Step size schedule $\{\alpha_{\max}, \alpha_1, \ldots \alpha_{\min}\}$, $\beta_{\max}$, $\beta_{\min}$, number of proposals per step $t = 10$, initial state $\theta$, target acceptance rate $\rho^*$
1: $\beta_{\text{floor}} = \beta_{\min}$, $\beta_{\text{ceil}} = \beta_{\max}$
2: $\beta$-sched $\leftarrow \{\beta_{\max}\}$
3: **for** $i$ in $\{1, 2, \ldots s - 1\}$ **do**
4:     proposed-params $\leftarrow$ LinSpace($\beta_{\text{floor}}$, $\beta_{\max}$, $t$) $\triangleright$ *Create $t$ potential balance parameters for index $i$ in the schedule*
5:     initialize bookkeeping to keep track of proposed states and acceptance rates
6:     **for** $\beta \in$ proposed-params **do**
7:         $\theta' \sim Q_{\alpha_i, \beta}(\cdot | \theta)$ $\triangleright$ *Use current proposed $\beta$ to take a sampling step*
8:         $\rho = A(\theta' | \theta, \alpha_i, \beta)$ $\triangleright$ *Evaluate the acceptance rate of proposed $\beta$ for current $\alpha_i$*
9:         bookkeeping[$\beta$] $\leftarrow \theta', \rho$
10:     **end for**
11:     pick $\beta_i$ as $\beta \in$ bookkeeping largest $\rho$
12:     $\beta_{\text{ceil}} \leftarrow \beta_i$ $\triangleright$ *Shrink the range of potential balancing parameters by using assumption $\beta_i > \beta_{i+1}$*
13:     $\theta = \theta'$ corresponding to $\beta_i$
14: **end for**
15: $\beta$-sched.append($\beta_{\min}$)
16: return $\beta$-sched

---

Instead of resetting the chain each training iteration, we maintain a buffer of the generated samples that we use to calculate the second expectation. This method relies on the intuition that the model distribution does not vary too widely within one iteration. Using the intuition provided by [Du and Mordatch, 2019], we can view this process as updating the model parameters $\phi$ to put more weight on true samples and less weight on fake samples. By doing so, the model will in turn generate samples that closer to those from the true distribution.

### B.2 Persistent Contrastive Divergence with ACS

**Main Idea**    We can apply the ACS algorithm combining the automatic tuning of the cyclical schedule with the original PCD learning algorithm. Our goal in doing so is to improve PCD through better characterization of the entire model distribution. During training, we can view PCD as adjusting the model parameters to "push down" the probability of samples from the model distribution while "pushing up" samples from the true data distribution. Because our sampling method is able to explore the model's distribution more effectively than other samplers, we can adjust more regions of the model distribution at a quicker rate than previous sampling methods, which should improve the quality of gradient updates and thus lead to better model parameters. We adapt ACS to work within PCD by having the step size depend on the training iteration as opposed to the sampling iteration, with the corresponding $\alpha, \beta$ pair being used for all the sampling steps within the iteration. We include the complete learning algorithm in Algorithm 6.

**Cyclical Scheduling**    We find that the learning task requires a different approach to the cyclical scheduling than the sampling task. Rather than having a relative equal amount of exploration and exploitation, we find that it is more effective to use a cyclical schedule biased towards exploitation. However, exploration is still important as it enables the model to better represent the distribution as a whole rather than a few local modes. Given this, we construct a cyclical schedule consisting of one iteration that uses $\alpha_{\max}, \beta_{\max}$ with the rest using $\alpha_{\min}, \beta_{\min}$.

**Tuning**    One of the advantages of using the simplified cyclical schedule is that it only requires two pairs of hyper-parameters to be optimized. Thus we can leverage the EstimateAlphaMax and EstimateAlphaMin algorithm to both tune the respective $\alpha, \beta$ pair while also updating the persistent buffer. Not only does this reduce the additional overhead of the tuning component, but it allows the hyper-parameters to adapt to the changing EBM distribution.

---
**Algorithm 6** ACS for Persistent Contrastive Divergence
---
**Require:** Number Iterations $N$, EBM $E_\phi$, data-loader $D$, sampler $Q$, small sampling steps $S_{\text{small}}$, big sampling steps $S_{\text{big}}$, initial buffer $X_f$, cycle length $s$, $\alpha_{\text{floor}}$, $\alpha_{\text{ceil}}$, adaptive learning rate $\zeta$, adaptive budget BUDGET

1: **while** iteration $i \leq N$ **do**
2:    **for** $X_t \sim D$ **do**
3:       cycle number $c = \text{floor}(\frac{i}{C})$
4:       **if** $c \mod K = 0$ **then**
5:          **if** $i \mod s = 0$ **then**
6:             $X_f, \alpha_{\max} \leftarrow$ EstAlphaMax($\alpha_{\text{ceil}}$, budget=BUDGET, learning-rate $=\gamma$)
7:          **else**
8:             $X_f, \alpha_{\min} \leftarrow$ EstAlphaMin($\alpha_{\text{floor}}$, budget=BUDGET, learning-rate=$\gamma$)
9:          **end if**
10:          Update Sampler Step Schedule ▷ *Update the buffer by running either the AlphaMax or AlphaMin estimation algorithm*
11:       **else**
12:          **if** $i \mod s = 0$ **then**
13:             $S = S_{\text{big}}$
14:             $\alpha = \alpha_{\max}, \beta = \beta_{\max}$ ▷ *Use the $\alpha, \beta$ pair that best enables exploration*
15:          **else**
16:             $S = S_{\text{small}}$
17:             $\alpha = \alpha_{\min}, \beta = \beta_{\min}$ ▷ *Use the $\alpha, \beta$ pair that best enables exploitation*
18:          **end if**
19:          Construct $Q = Q_{\alpha,\beta}(\cdot|X_f)$ using (4)
20:          **for** sampling step in range($S_{\text{big}}$) **do**
21:             $X \sim Q(\cdot|X_f)$
22:             **if** $i \mod s = 0$ **then**
23:                $X_f \leftarrow X$
24:                **continue** ▷ *If $i$ is the first step of the cycle, omit the MH correction*
25:             **end if**
26:             $X_f \leftarrow X$ with acceptance probability as calculated in (6)
27:          **end for**
28:       **end if**
29:       Calculate $\mathbb{E}_{x\sim p_\phi}[\nabla_\phi E_\phi(x)]$ using $X_f$
30:       Calculate $\mathbb{E}_{x\sim p_{\text{data}}}[\nabla_\phi E_\phi(x)]$ using $X_t$
31:       $\nabla\mathcal{L}(\phi) = \mathbb{E}_{x\sim p_\phi}[\nabla_\phi E_\phi(x)] - \mathbb{E}_{x\sim p_{\text{data}}}[\nabla_\phi E_\phi(x)]$ ▷ *Estimate the gradient of the Maximum-Likelihood objective as in* (12)
32:       $\phi = \phi - \gamma_\phi \nabla\mathcal{L}(\phi)$
33:       $i += 1$
34:    **end for**
35: **end while**

## C   Theoretical Results

We define the problem setting in more detail. We have a target that is of the form

$$\pi(\theta) = \frac{1}{Z} \exp(U(\theta)).$$

We consider the proposal kernel as

$$Q_{\alpha,\beta}(\theta'|\theta) \propto \exp\left\{\beta \nabla U(\theta)^T (\theta' - \theta) - \frac{1}{2\alpha}\|\theta' - \theta\|^2\right\}$$

and consider the transition kernel as

$$p(\theta' \mid \theta) = \left(\frac{\pi(\theta')Q_{\alpha,\beta}(\theta \mid \theta')}{\pi(\theta)Q_{\alpha,\beta}(\theta' \mid \theta)} \wedge 1\right) Q_{\alpha,\beta}(\theta' \mid \theta) + (1 - L(\theta))\, \delta_\theta(\theta')$$

where $\delta_\theta(\theta')$ is the Kronecker delta function and $L(\theta)$ is the total acceptance probability from the point $\theta$ with

$$L(\theta) = \sum_{\theta' \in \Theta} \left(\frac{\pi(\theta')Q_{\alpha,\beta}(\theta|\theta')}{\pi(\theta)Q_{\alpha,\beta}(\theta'|\theta)} \wedge 1\right) Q_{\alpha,\beta}(\theta'|\theta).$$

We also define

$$Z_{\alpha,\beta}(\theta) = \sum_{x \in \Theta} \exp\left\{\beta \nabla U(\theta)^T (x - \theta) - \frac{1}{2\alpha}\|x - \theta\|^2\right\}$$

which is the normalizing constant for the proposal kernel.

### C.1   Proof of Lemma 5.3

*Proof.* By including the balancing parameter, we start by noting that

$$Q_{\alpha,\beta}(\theta'|\theta) = \frac{\exp\left\{\beta\nabla U(\theta)^T (\theta' - \theta) - \frac{1}{2\alpha}\|\theta' - \theta\|^2\right\}}{\sum_{\theta \in \Theta} \exp\left\{\beta \nabla U(\theta)^T (\theta - \theta) - \frac{1}{2\alpha}\|\theta - \theta\|^2\right\}} \tag{14}$$

Consider the term,

$$\beta \nabla U(\theta)^T (\theta' - \theta) = \beta \left(-U(\theta) + U(\theta')\right) - \frac{\beta}{2}(\theta - \theta')^T(\int_0^1 \nabla^2 U((1-s)\theta + s\theta')\, ds)(\theta - \theta') \tag{15}$$

Substituting (15) in (14), the numerator of $Q_{\alpha,\beta}(\theta, \theta')$

$$\begin{aligned}
\beta\nabla U(\theta)^T (\theta' - \theta) - \frac{1}{2\alpha}\|\theta' - \theta\|^2 =&\, \beta\left(-U(\theta) + U(\theta')\right) \\
&- \frac{\beta}{2}(\theta - \theta')^T \left(\int_0^1 \nabla^2 U((1-s)\theta + s\theta')\, ds\right)(\theta - \theta') \\
&- \frac{1}{2\alpha}(\theta - \theta')^T I(\theta - \theta') \\
=&\, \beta\left(-U(\theta) + U(\theta')\right) \\
&- \frac{1}{2}(\theta - \theta')^T \left(\beta\int_0^1 \nabla^2 U((1-s)\theta + s\theta')\, ds + \frac{1}{\alpha}I\right)(\theta - \theta')
\end{aligned}$$

.

From Assumption 5.1 (U is $M$-gradient Lipschitz), we have

$$\beta \int_0^1 \nabla^2 U((1-s)\theta + s\theta')\, ds)(\theta - \theta') + \frac{1}{\alpha}I \geq \left(\frac{1}{\alpha} - \beta M\right) I$$

Since $\alpha < 1/\beta M$, the matrix $\left(\frac{1}{2\alpha} - \beta M\right) I$ is positive definite. We note that

$$p(\theta'|\theta) = \left(\frac{\pi(\theta')Q_{\alpha,\beta}(\theta|\theta')}{\pi(\theta)Q_{\alpha,\beta}(\theta'|\theta)} \wedge 1\right) Q_{\alpha,\beta}(\theta'|\theta) + (1 - L(\theta))\,\delta_\theta(\theta') \tag{16}$$

$$\geq \left(\frac{\pi(\theta')Q_{\alpha,\beta}(\theta|\theta')}{\pi(\theta)Q_{\alpha,\beta}(\theta'|\theta)} \wedge 1\right) Q_{\alpha,\beta}(\theta'|\theta) \tag{17}$$

$$= \left(\frac{Z_{\alpha,\beta}(\theta)}{Z_{\alpha,\beta}(\theta')} \wedge 1\right) Q_{\alpha,\beta}(\theta'|\theta). \tag{18}$$

$$Z_{\alpha,\beta}(\theta) = \sum_{x \in \Theta} \exp\left\{\beta\,\nabla U(\theta)^T (x - \theta) - \frac{1}{2\alpha}\|x - \theta\|^2\right\}$$

$$= \sum_{x \in \Theta} \exp\left\{-\beta\,(U(\theta) - U(x)) - \frac{1}{2}(\theta - x)^T (\beta \int_0^1 \nabla^2 U((1-s)\theta + sx)\,ds)(\theta - x) + \frac{1}{\alpha}I)(\theta - x)\right\}.$$

This can be seen as

$$\pi(\theta)Q_{\alpha,\beta}(\theta'|\theta) = \frac{1}{Z\,Z_{\alpha,\beta}(\theta)} \exp\left\{\beta\,(U(\theta) + U(\theta')) - (\theta' - \theta)^T \left(\frac{1}{2\alpha}I + \frac{\beta}{2}\int_0^1 \nabla^2 U((1-s)\theta + s\theta')ds\right)(\theta' - \theta)\right\}.$$

Since Assumption 5.2 holds true in this setting, we have an $m > 0$ such that for any $\theta \in conv(\Theta)$

$$-\nabla^2 U(\theta) \geq m\,I.$$

From this, one notes that

$$\exp\left(-\beta U(\theta) - \frac{1}{2}\left(\frac{1}{\alpha} - \beta\,m\right) diam(\Theta)^2\right) \sum_{x \in \Theta} \exp\left(\beta U(x)\right) \leq Z_{\alpha,\beta}(\theta) \leq \exp\left(-\beta U(\theta)\right) \sum_{x \in \Theta} \exp\left(\beta U(x)\right)$$

where the right-hand side follows from the fact that $\alpha < 1/(\beta M)$. Therefore,

$$\frac{Z_{\alpha,\beta}(\theta)}{Z_{\alpha,\beta}(\theta')} \geq \frac{\exp\left\{\beta\,(-U(\theta) + U(\theta'))\right\}}{\exp\left\{\frac{1}{2}\left(\frac{1}{\alpha} - \beta m\right) diam(\Theta)^2\right\}}$$

Also note that

$$Q_{\alpha,\beta}(\theta'|\theta) = \frac{\exp\left\{\beta\,(-U(\theta) + U(\theta')) - (\theta - \theta')^T \left(\frac{1}{2\alpha}I + \frac{\beta}{2}\int_0^1 \nabla^2 U((1-s)\theta + s\theta')\right)(\theta - \theta')\right\}}{\sum_{\theta' \in \Theta} \exp\left\{\beta\,(-U(\theta) + U(\theta')) - (\theta - \theta')^T \left(\frac{1}{2\alpha}I + \frac{\beta}{2}\int_0^1 \nabla^2 U((1-s)\theta + s\theta')\right)(\theta - \theta')\right\}}$$

$$\geq \frac{\exp\left\{\beta\,\langle\nabla U(\theta), \theta' - \theta\rangle - \frac{1}{2\alpha}\|\theta - \theta'\|^2\right\}}{\sum_{\theta'\Theta} \exp\left\{\beta\,(-U(\theta) + U(\theta'))\right\}}.$$

We also note that

$$-\beta\,\langle\nabla U(\theta), \theta' - \theta\rangle + \frac{1}{2\alpha}\|\theta - \theta'\|^2 = \beta\,\langle-\nabla U(\theta) + \nabla U(a), \theta' - \theta\rangle + \beta\,\langle-\nabla U(a), \theta' - \theta\rangle + \frac{1}{2\alpha}\|\theta - \theta'\|^2$$

$$\leq \beta\,\langle-\nabla U(\theta) + \nabla U(a), \theta' - \theta\rangle + \beta\,\langle-\nabla U(a), \theta' - \theta\rangle + \frac{1}{2\alpha}diam(\Theta)^2$$

$$\leq \beta\,\|-\nabla U(\theta) + \nabla U(a)\|\|\theta' - \theta\| + \beta\,\|\nabla U(a)\|\|\theta' - \theta\| + \frac{1}{2\alpha}diam(\Theta)^2$$

$$\leq \beta\,\|-\nabla U(\theta) + \nabla U(a)\|diam(\Theta) + \beta\|\nabla U(a)\|diam(\Theta) + \frac{1}{2\alpha}diam(\Theta)^2$$

$$\leq \left(\beta M + \frac{1}{2\alpha}\right) diam(\Theta)^2 + \beta\|\nabla U(a)\|\,diam(\Theta).$$

Combining, we get

$$p(\theta'|\theta) \geq \epsilon_{\beta,\alpha} \frac{\exp\left\{\beta U(\theta')\right\}}{\sum_{\theta'\Theta} \exp\left\{\beta U(\theta')\right\}}$$

where

$$\epsilon_{\beta,\alpha} = \exp\left\{-\left(\frac{1}{\alpha} + \beta M - \frac{\beta\,m}{2}\right) diam(\Theta)^2 - \|\nabla U(a)\|\,diam(\Theta)\right\}.$$

$\square$

## C.2 Proofs of Proposition C.1 and Corollary C.2

**Proposition C.1.** *Let $P_1, P_2, \cdots P_s$ be Markov transition operators with kernels $p_1, p_2, \cdots p_s$ with respect to a reference measure $\eta$. Also, let $p_i(\theta'|\theta) \geq \epsilon_i \nu_i(\theta')$ for some density $\nu_i$ on $\Theta$ and $\epsilon_i > 0$ with respect to a reference measure $\eta$. Then, for the Markov operator $\hat{P}_i$ defined with respect to the kernel as*

$$\hat{p}_i(\theta'|\theta) = \int_{\Theta^{S-1}} p_{i+1}(\theta_1|\theta) p_{i+2}(\theta_2|\theta_1) \cdots p_s(\theta_{s-i+1}|\theta_{s-i})$$
$$\cdots p_i(\theta'|\theta_{s-1}) d\eta(\theta_1) d\eta(\theta_2) \cdots d\eta(\theta_{s-1}),$$

*we have*

$$\hat{p}_i(\theta'|\theta) \geq \epsilon_i \nu_i(\theta'), \forall \theta \in \Theta \cdot$$

*Proof.* The proof is straightforward by using the minorization of $p_i$. Indeed, one has

$$\hat{p}(\theta'|\theta) = \int_{\Theta^{S-1}} p_{i+1}(\theta_1|\theta) p_{i+2}(\theta_2|\theta_1) \cdots p_s(\theta_{s-i+1}|\theta_{s-i}) \cdots p_i(\theta'|\theta_{s-1}) d\eta(\theta_1) d\eta(\theta_2) \cdots d\eta(\theta_{s-1})$$

$$\geq \epsilon_i \nu_i(\theta') \int_{\Theta^{S-1}} p_{i+1}(\theta_1|\theta) p_{i+2}(\theta_2|\theta_1) \cdots p_s(\theta_{s-i+1}|\theta_{s-i}) \cdots p_{i-1}(\theta_{s-1}|\theta_{s-2}) d\eta(\theta_1) \cdots d\eta(\theta_{s-1})$$

$$\geq \epsilon_i \nu_i(\theta')$$

which establishes the result. $\qquad\square$

Note that in Algorithm 1, for each cycle, we go through s steps corresponding to the step size and balancing parameter schedules ($\{\alpha_1, \alpha_2, \cdots \alpha_s\}$) and ($\{\beta_1, \beta_2, \cdots \beta_s\}$). Let $P_1, P_2, \cdots, P_s$ be the Markov operators corresponding to them.

**Corollary C.2.** *Let Assumptions 5.1 and 5.2 hold. Then*

$$P_1 P_2 P_3 \cdots P_s(\theta, A) \geq \epsilon_s \nu_s(A)$$

*for any measurable subset $A$ of $\Theta$.*

*Proof.* The proof is immediate from Proposition C.1. $\qquad\square$

## C.3 Additional Lemma

**Lemma C.3.** *Let Assumption 5.2 hold with $\Theta$ compact. Then, there exists some $m > 0$ such that for any $\theta \in conv(\Theta)$, $\lambda_{\min}(\nabla^2 - U(\theta)) > m$.*

*Proof.* Note that since $\Theta$ is compact $conv(\Theta)$ is also compact. This is easy to see as we only need to establish that $conv(\Theta)$ is closed and bounded by the Heine-Borel Theorem. Take any element in $\theta \in conv(\Theta)$. By definition, $\theta = \alpha\theta_1 + (1-\alpha)\theta_2$ for some $\theta_1, \theta_2 \in \Theta$ and $0 \leq \alpha \leq 1$. Since $\Theta$ is compact, we know that there exists $M > 0$ such that $\|\theta_i\| < M$ for $i = 1, 2$. Therefore $\|\theta\| < M$ by triangle inequality. Thus the set is bounded. The fact that it is closed is also easy to see. Take any sequence $x_n$ in $conv(\Theta)$. This implies there exists $\alpha_n, \theta_{1,n}, \theta_{2,n}$ such that $x_n = \alpha_n \theta_{1,n} + (1-\alpha_n)\theta_{2,n}$. Since $x_n$ converges as our assumption, it is Cauchy which in turn implies each of $\alpha_n, \theta_{1,n}, \theta_{2,n}$ is Cauchy as $\Theta$ is bounded. Thus the proof immediately follows. Now, consider each $\theta \in conv(\Theta)$. There exits a $B(\theta, r_\theta)$ such that $\nabla^2 - U(\theta') \geq m_\theta I$ for all $\theta' \in B(\theta, r_\theta)$. Since $conv(\Theta) \subset \cup_{\theta \in \Theta} B(\theta, r_\theta)$, this is an open cover of $conv(\Theta)$. Since $conv(\Theta)$ is compact, there exists $\theta_1, \theta_2, \cdots, \theta_k$ such that $conv(\Theta) \subset \cup_{i=1}^k B(\theta_i, r_{\theta_i})$. Thus for each $i$ we have $\nabla^2 - U(\theta') \geq m_{\theta_i} I$ when $\theta \in B(\theta_i, r_{\theta_i})$. Thus $\nabla^2 - U(\theta) \geq \min_{1 \leq i \leq k} m_{\theta_i} I$ for all $\theta \in conv(\Theta)$. Hence we are done. $\qquad\square$

# D  Additional Experimental Results and Details

Here, we include the full details for all the experiments we include in this paper, as well as some additional results. All experiments were run on a single RTX A6000.

## D.1 Multi-modal Experiment Design

**Synthetic Distribution** In order to construct a distribution that is easy to visualize, we first must define a few experiment parameters. We must define the space between the modes, the total number of modes, and the variance of each mode $\sigma$. For convenience, we have the number of modes as 25, which is a perfect square. We define the space between modes as 75, and the variance for each mode $\sigma^2$ as .15. Given this, we can calculate the maximum value for each coordinate as follows:

$$\text{MaxVal} = (\sqrt{\text{NumModes}} + 1) * \text{SpaceBetweenModes}$$

We can calculate the coordinate value for each mode $\mu_{i,j}$ as follows:

$$\mu_{i,j}[0] = \frac{\text{MaxVal}}{\sqrt{\text{NumModes}} + 2}(i + 1)$$
$$\mu_{i,j}[1] = \frac{\text{MaxVal}}{\sqrt{\text{NumModes}} + 2}(j + 1)$$

**Sampler Configuration** Our goal in this experiment is to demonstrate how gradient-based samplers typically behave when faced with a distribution with modes that are far apart. In order for this experiment to be meaningful, it is important that the representation of each sample respect the notion of distance between the integer values. For this reason, we cannot use a categorical distribution or represent each coordinate with a one-hot encoding, as every sample in this representation would be within a 2-hamming ball of every other point.

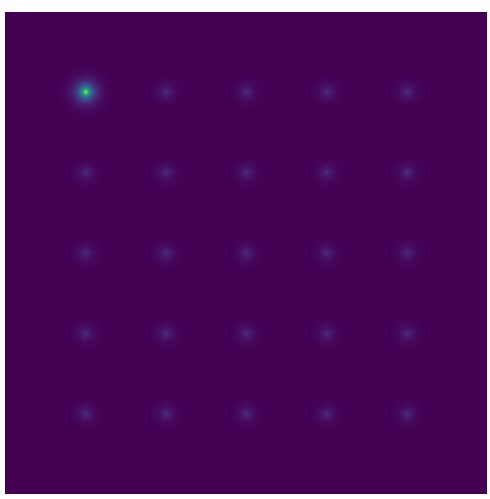

In order to determine the step sizes for the baselines, we tune each until we reach an acceptance rate around .574. For DMALA, this ends up being around $\alpha = 53$. For the any-scale sampler, we set the initial step size to be the same and use their implemented adaptive algorithm.

For the cyclical sampler, we set $\alpha_{\max} = 1575$, $\alpha_{\min} = 3$, and steps per cycle $s = 20$. Because the goal of the experiment is to demonstrate the need for larger step sizes along with smaller step sizes, we do not use the automatic tuning algorithm on this example as restricting the space to be ordinal changes the optimal setting for $\alpha_{\text{ceil}}$. In most practical cases, the samples would be represented by a categorical or binary form, which the proposed tuning algorithm is able to handle as demonstrated by the performance on real data distributions.

Figure 5: Uneven multi-modal target distribution. While the top-left mode does have the most mass, only sampling from this mode will result in an inaccurate representation of the target distribution.

**Uneven Multi-modal Distributions** Not only does a cyclical step-size enable more accurate sampling in highly multi-modal distributions, but it is also able to handle distributions where the modes are weighted unevenly. This problem is more difficult since this requires not only exploring all the modes of a distribution, but ensuring that the less likely modes are not over represented in the generated samples. We provide a visual comparison between the target distribution, the estimated distribution from DMALA, and the estimated distribution from ACS in Figure 5.

Since the modes may not be clear due to the nature of this specific problem, we also include a quantitative comparison between DMALA and ACS in Table 3 by computing the KL divergence between the estimated distribution and the target distribution in addition to the average energy of the generated samples. Through both Table 3 and Figure 5, we observe that a cyclical step-size enables accurate sampling from uneven multi-modal distributions.

Furthermore, it is interesting to observe that generating more high-probability samples does not necessarily correspond to accurate sampling, highlighting the difference in goals between generating very likely samples and accurately sampling the target distribution.

Table 3: Quantitative comparison of sampler performance on uneven multi-modal distribution. ACS retains the ability to accurately capture all the modes within the distribution despite the uneven weighting.

|  | DMALA | ACS |
|---|---|---|
| KL Divergence | 0.70 | 0.13 |
| Average Energy | $-2.66 \pm 1.68$ | $-3.39 \pm 1.63$ |

## D.2   RBM Sampling

**RBM Overview**   We will give a brief overview of the Block-Gibbs sampler used to represent the ground truth of the RBM distribution. For a more in-depth explanation, see Grathwohl et al. [2021]. Given the hidden units $h$ and the sample $x$, we define the RBM distribution as follows:

$$\log p(x, h) = h^T W x + b^T x + c^T - \log Z \qquad (19)$$

As before, Z is the normalizing constant for the distribution. The sample $x$ is represented by the visible layer with units corresponding to the sample space dimension and $h$ represents the model capacity. It can be shown that the marginal distributions are as follows:

$$p(x|h) = \text{Bernoulli}(Wx + c)$$
$$p(h|x) = \text{Bernoulli}(W^t h + b)$$

The Block-Gibbs sampler updates $x$ and $h$ alternatively, allowing for many of the coordinates to get changed at the same time, due to utilizing the specific structure of the RBM model.

**Experiment Setup**   Similar to the experimental setup of Zhang et al. [2022a], we use RBM models with 500 hidden units and 784 visible units. We adopt the same training protocol, except we train the RBM with 100 steps of Contrastive Divergence as opposed to 10. We also train the models for 1000 iterations as opposed to a single pass through the dataset. We find that this enables the RBMs to generate more realistic samples. We include the generated images in Figure 6 to demonstrate that these models have learned the dataset reasonably well.

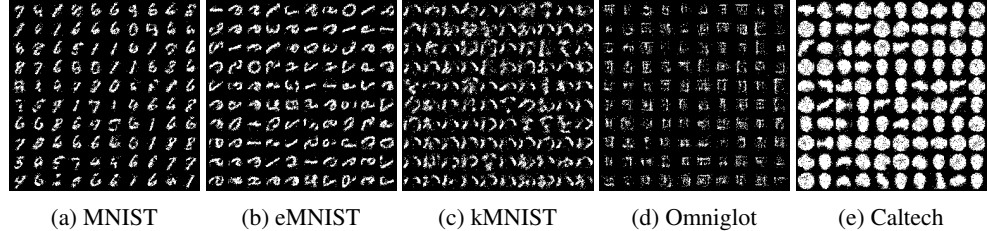

|  (a) MNIST  |  (b) eMNIST  |  (c) kMNIST  |  (d) Omniglot  |  (e) Caltech  |

Figure 6: Images sampled from RBMs trained by Contrastive-Divergence with Block Gibbs. We use Block Gibbs as the sampling algorithm to produce these images as well.

**Sampler Configuration**   For GWG, we use the same settings as Grathwohl et al. [2021], for DMALA, we set step size to .2, and for AB we use the default hyper-parameters for the first order sampler.

For ACS, we use $\rho^* = .5, \beta_{\max} = .95, \zeta = .5$, cycle length $s = 20$ for all the datasets. We also fix the total overhead of the tuning algorithm to 10% of the total sampling steps.

**Escape from Local Modes**   In addition to using the same initialization as Zhang et al. [2022a], Grathwohl et al. [2021], we extend the experiment to measure the ability of a sampler to escape from local modes. We initialize the sampler within the most likely mode, as measured by unnormalized energy of the RBM. Samplers that are less prone to getting trapped in local modes will be able to converge quicker to the ground truth, as measured by log MMD. We include the performance of the various samplers across 11 random seeds in 7. ACS demonstrates superior robustness to mode-specific initialization due to its capability to escape from local modes.

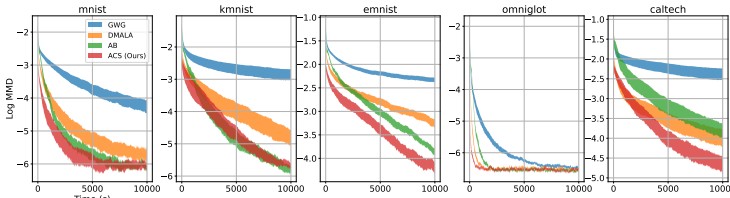

Figure 7: Log MMDs v.s sampling iteration across various datasets. ACS demonstrates more robust sampling behavior across the datasets than other methods, as evidenced by superior convergence on all datasets except KMNIST. We do note that ACS performance is still competitive on KMNIST with the added benefit of a smaller standard error.

**Generated Images**    We found that a visual inspection of the generated images demonstrates the ability of ACS to escape local modes. We include the generated images in Figure 8.

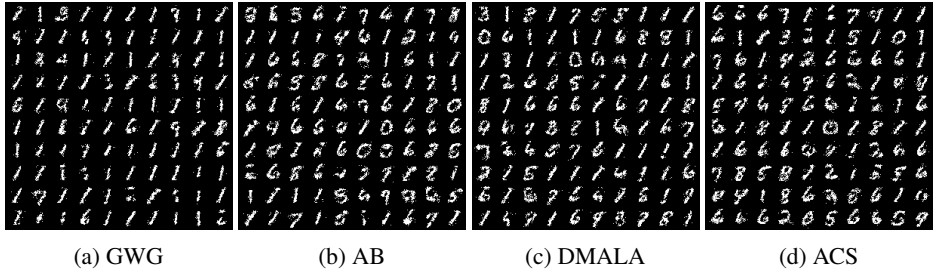

(a) GWG            (b) AB            (c) DMALA            (d) ACS

Figure 8: Images sampled from RBM trained on MNIST when the sampler is initialized to most likely mode. ACS is able to generate a diverse range of digits, demonstrating its ability to escape from modes. It should also noted that while AB is able to generate a diverse range of digits as well, the images are slightly less clear than those generated by ACS.

We can make two primary inferences from the generated images: the first being that ACS is able to escape from local modes and explore the distribution as a whole, as demonstrated by the wide range of generated images; and that ACS does not compromise on the ability to characterize each mode as evidenced by the quality of generated samples.

**Sampling Speed**    While the run time can vary depending on the specific implementation of a given sampling algorithm, we illustrate the efficiency of ACS in Figure 9. ACS is able to outperform DMALA in terms of convergence with respect to time, even including the overhead of the tuning algorithm.

### D.3    EBM Sampling

**Base EBM Training**    In order to train the EBMs, we use Gibbs-with-Gradient to sample the EBM distribution during PCD, following the same training protocol as Grathwohl et al. [2021]. We train these models for 50,000 iterations total with 40 sampling steps per iteration and use the parameters corresponding to the best log likelihood scores on the validation dataset.

**Experimental Design**    For each of the trained models, we evaluate the samplers based on how quickly the average energy of the generated samples rises. This gives an estimate of the speed at which a sampler is able to reach a stationary distribution.

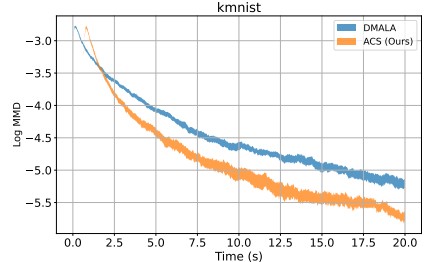

Figure 9: Log MMD for the RBM sampling task across time for both DMALA and ACS for the kMNIST dataset. We observe that while ACS is offset slightly in the beginning due to the initial tuning algorithm, it is quickly able to demonstrate superior convergence.

**Sampler Configuration**    For GWG, we use the same settings as Grathwohl et al. [2021], for DMALA, we set step size to .2, and for AB we use the default hyper-parameters for the diagonal variant of the AB sampler. We choose this variant as this is what they evaluate for their experiments when measuring mixing speed of samplers on EBMs.

For ACS, we use $\rho^* = .5, \beta_{\max} = .8, \zeta = .5$, cycle length $s = 20$ for all the datasets. As in RBM Sampling, we fix the total overhead of the tuning algorithm to 10% of the total sampling steps.

**Sampler Performance**    It is worth commenting on the similarity in performance between ACS and DMALA when sampling from Caltech. We find that when sampling from the EBM trained on the Caltech dataset, ACS finds a $\alpha_{\max}$ similar to $\alpha_{\min}$, thus making the ACS sampler similar to DMALA for this specific case. We hypothesize that small step sizes are most effective for this dataset. The results in Figure 3 demonstrate that ACS can handle such cases automatically: while the step size for DMALA must be hand-tuned, the ACS method can automatically adapt to a suitable step size schedule.

**Generated Images**    We include the images generated by ACS when sampling from deep EBMs in Figure 10.

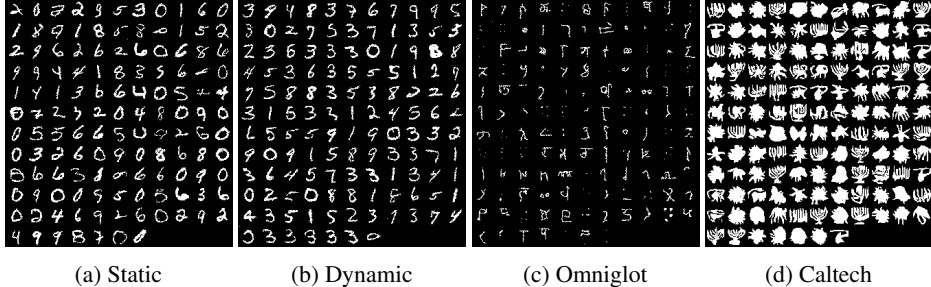

(a) Static          (b) Dynamic          (c) Omniglot          (d) Caltech

Figure 10: Generated Images from applying ACS sampling to deep EBMs trained with GWG. These samples capture multiple different modes while retaining good sample quality, demonstrating the benefit of our ACS method.

### D.4    RBM Learning

**Experiment Design**    We use the same RBM structure as the sampling task, with 500 hidden units and 784 visible units. However, we apply the samplers of interest to the PCD algorithm introduced by Tieleman [2008]. The model parameters are tuned via the Adam optimizer with a learning rate of .001.

In order to evaluate the learned RBMs, we run AIS with Block-Gibbs as the sampler to calculate the log likelihood values for the models Neal [2001]. We run AIS for 100,000 steps, which is adequate given the efficiency of Block Gibbs for this specific model.

**Sampler Configuration**    For DMALA, we use a step size of .2. For the ACS algorithm, we set $\beta_{\max} = .9, \rho^* = .5$ for all the data-sets. We do modify the number of cycles for each data-set as different distributions require different amounts of exploration and exploitation. We use cycle length of 8 for MNIST, eMNIST, and kMNIST; we use 20 for Omniglot and Caltech silhouettes. This difference reflects the specific needs for each dataset in terms of exploration and exploitation – more complex datasets tend to need longer cycles in order to better exploit each region, while simpler datasets tend to need shorter cycles in order to capture all the modes of the learned distribution. In Figure 11, we show the samples generated from AIS for 100,000 steps as opposed to the persistent buffer as this forms a longer MCMC chain, thus giving a better visual of what the learned distribution represents.

In order to ensure that the overhead for the tuning algorithm does not add to the overall computational cost, we spread out the computations of the EstimateAlphaMin algorithm throughout the training process. We keep a running list of $\alpha_{\min}$ and set $\alpha_{\text{floor}}$ to be one standard deviation below the mean of this list. By doing this, we start closer to what the ideal $\alpha_{\min}$. For EstimateAlphaMax, we simply call

the tuning function every 50 cycles containing 50 training iterations, with $\alpha_{ceil} = 5$. As the initial step does not use the Metropolis-Hastings correction and has half the sampling steps, the budget for each call of EstimateAlphaMax can be seen as coming in part by the computation saved.

**Results**   We include the AIS results for RBMs trained with different sampling methods in Table 4. We see that ACS achieves superior log likelihood results when compared to other sampling methods across all datasets. Furthermore, the AIS results are consistently close to those achieved by Block-Gibbs, which can be considered close to ideal for RBMs since it leverages the known structure of the model.

**Generated Images**   We include the generated images from the RBMs trained using different samplers in Figure 11.

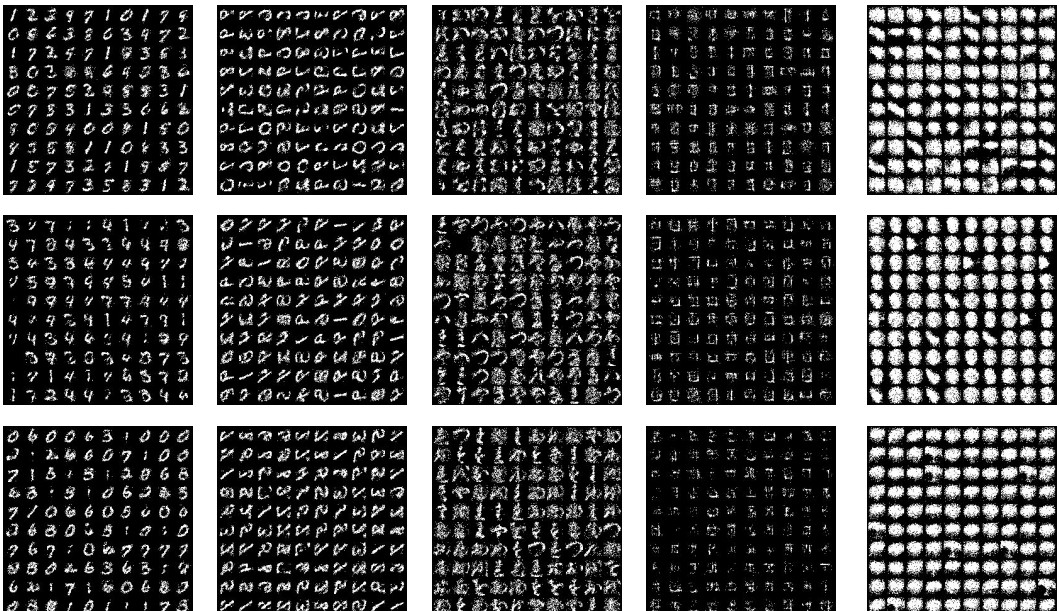

Figure 11: Generated images from RBMs trained with different samplers. First row corresponds to GWG, second row corresponds to DMALA, and final row corresponds to ACS. First column represents models trained on MNIST, second on eMNIST, third on kMNIST, fourth on Omniglot, and fifth on Caltech Silhouettes. Images are generated via AIS for 100,000 steps.

In general, the images generated from the ACS-trained RBM capture more modes than other methods, except for the Caltech Silhouettes dataset. In particular, all the methods struggle to generate reasonable images for this dataset. We hypothesize that this is due to the increased complexity of the distribution relative to the other datasets – Caltech Silhouettes is composed of the silhouettes from real objects, whereas the other datasets are hand-written symbols. This hypothesis is supported by the generated images in Figure 6, where the

Table 4: Log likelihood scores for RBM learning on test data as estimated by AIS. ACS outperforms all gradient-based baselines across all datasets.

|  | GB | GWG | DMALA | ACS |
|---|---|---|---|---|
| MNIST | *-191.98* | -387.34 | -278.35 | **-249.55** |
| eMNIST | *-317.78* | -590.97 | -324.34 | **-304.96** |
| kMNIST | *-357.69* | -681.28 | -436.3538 | **-407.39** |
| Omniglot | *-161.73* | -276.81 | -222.61 | **-220.71** |
| Caltech | *-511.65* | -827.45 | -427.29 | **-396.04** |

images generated when using Block-Gibbs on Caltech Silhouettes also seem less reasonable than the samples obtained from different datasets. Since Block-Gibbs is the best sampler for this specific model as it leverages the known structure of the RBM, this appears to be unavoidable as a result of limited model capacity. This motivates our experiments with deep convolutional EBMs, where we can understand how our method does when using a model architecture with sufficient capacity.

### D.5 EBM Learning

**Experiment Design**   We use the same EBM model architecture as Zhang et al. [2022a], Grathwohl et al. [2021] and follow the same experimental design, with the only change being to the number of sampling steps alotted for each sampler.

In order to determine the number of sampling steps that we could use for ACS-PCD, we tested different sampling steps. For Static/Dynamic MNIST and Omniglot, we found that we only needed to use 10 sampling steps to achieve good models. However, we observed divergence when training models on Caltech. In order to determine what number of sampling steps to use for ACS, we do a grid search over the number of sampling steps and $\rho^*$ with all other values remaining the same. We test sampling steps of 10, 20, and 30; and we use $\rho^*$ .5, .6, .7, .8. We decide which hyper-parameters to use based on when training diverged the latest; and we use the best model parameters as indicated by validation log likelihood. We use 10 sampling steps for Static, Dynamic MNIST, and Omniglot, while we found 30 was the minimum we could use for Caltech Silhouettes and obtain reasonable results. We apply this number of sampling steps for both DMALA and ACS to demonstrate how the methods compare when facing a similar budget constraint.

In order to evaluate these learned models, we use the same evaluation protocol as Zhang et al. [2022a], Grathwohl et al. [2021]. We run AIS for 300,000 iterations using Gibbs-With-Gradient as the evaluation sampler. By following the same experimental design as previous works, we can draw meaningful comparisons from previous results in Grathwohl et al. [2021].

**Sampler Configuration**   For DMALA, we use a step size of .15 as used in Zhang et al. [2022b]. For ACS, we use 200 sampling steps for EstimateAlphaMax and EstimateAlphaMin. For Static MNIST, Dynamic MNIST, and Omniglot, we set the algorithm to tune $\alpha_{max}$ and $\alpha_{min}$ every 25 cycles, where each cycle has 50 training iterations. The additional overhead of this is 16,000 extra sampling steps, which is a 3.2% of the total budget of 500,000 sampling steps. For Caltech Silhouettes, we have to adapt every 10 cycles with the same number of training iterations. This results in 40,000 additional sampling steps due to the tuning algorithm. For this specific dataset, because we use 30 sampling steps, the additional cost is 2.6% of the total sampling steps 1,500,000.

In terms of the final parameters for cycle length and sampling steps, we find that we can use the same $\rho^*$ across all datasets, with the exception of Caltech Silhouettes. For Static/Dynamic MNIST and Omniglot, we were able to use $\rho^* = .5$ and For this dataset, we found good results by setting $\rho^* = .7$. We hypothesize that the need for a higher acceptance rate is due to the fundamental difference between Caltech Silhouettes and the other datasets, as previously mentioned. Because Caltech Silhouettes contain samples are derived from real objects, they are more complex than the hand-written figures.

**Experimental Results**   In addition to the empirical results in Table 1, we provide some qualitative data in the form of the generated images from the PCD buffer when using ACS. We choose to include the buffer images for this experiment as the chain from the persistent buffer is much longer than the chain from AIS due to the increased training duration: the chain from AIS is obtained using 300,000 sampling steps whereas the persistent buffer is obtained from 500,000 sampling steps on Static/Dynamic MNIST and Omniglot, 1,500,000 sampling steps for Caltech Silhouettes. By visualizing the generated images from the longer chain, we get a better understanding of the quality of the trained distribution. We put the images in Figure 12.

We also observe that this behavior is not unique to ACS and does occur when Gibbs-With-Gradient and DMALA are used with 40 sampling steps as indicated. Instability is common when training deep EBMs, and this is most likely why the original experimental design included check-pointing throughout the training process as well as comparisons based on the validation set. We also note that despite this behavior, the trained models are able to generate fairly realistic images. We present the images from the PCD buffer for ACS below in figure.

When the images in Figure 12 are taken in context of the improvements in log likelihoods as presented in Table 1, the results indicate the benefits of using ACS when learning multi-modal discrete distributions.

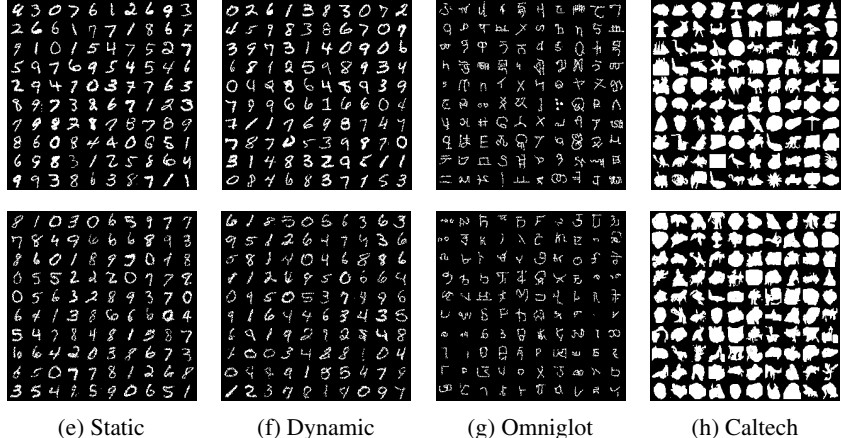

| (e) Static | (f) Dynamic | (g) Omniglot | (h) Caltech |

Figure 12: The example images from the representative datasets, along with the samples generated from the persistent buffer when using ACS as the sampler for PCD. The images on the top row are examples from the dataset, while the bottom row are from the trained EBM. The images generated from ACS are remarkably similar to those from the dataset, demonstrating that the model is capable of generating high-quality samples.

### D.6 Text Infilling

**Experimental Design** For both datasets, we sample 100 sentences randomly and mask $50\%$ of the tokens. We use a pretrained RoBERTa model available through the Hugging Face API Liu et al. [2019]. We run 25 separate chains for each example and take the final state of each chain to be a sample. We then take the top-5 most likely samples and use these for empirical comparisons.

We define the energy function the same as in Zhang et al. [2022a]. Let us define a sentence of length $d$ $\theta = \{\theta_1, \theta_2, \ldots \theta_d\}$, where $\theta_i$ is a one hot vector over vocabulary $V$. Let $M \subset \{1, 2, \ldots d\}$ the set of indices we wish to sample. We define the function $f(\theta_i|\theta_{\neg i})$ to be the log probability distribution over $V$ for the $i$ position conditioned on all other positions. Given this, we define the energy function for the sentence $\theta$ to be as follows:

$$U(\theta) = \sum_{m \in M} f(\theta_m|\theta_{\neg m}) \tag{20}$$

**Sampler Configuration** For DMALA, we tune the step-size to achieve an acceptance rate of 50%, which ends up being $\alpha = .5$. For ACS, we use a cycle length of 20. We use the same hyper-parameters for the tuning algorithms as in previous tasks, demonstrating that our algorithm can be applied across domains and tasks with little modification. We include example generations for both ACS and DMALA in Figure 13.

**Perplexity v.s Sampling Accuracy** In Table 2, we observe that the ACS generations have higher perplexity than the DMALA generations. While perplexity is a popular means of evaluating language generations, it is important to recognize that perplexity is based on the likelihood of the generation under the language model. This metric is biased towards frequent patterns and does not account for diverse modes. Therefore, it does not completely align with the goal of MCMC, which is to accurately characterize the target distribution.

Minimizing the average perplexity of the sample corresponds to maximizing the average likelihood of the generations, which can be at odds with the goal of accurately capturing the target distribution. We illustrate this in Appendix D.1, where we compare the performance of DMALA and ACS when sampling from a uneven multi-modal distribution.

- The comedy that follows feels hackneyed or just plain crude, calculated to provoke shocked stares, without opening up to a deeper truth.

- This comedy could either be hacky, or just plain crude, calculated to provoke shocked curiosity, without opening up to a deeper insight.

- A comedy that feels slightly hacky, or just plain crude, calculated to achieve shocked results, without coming up with a deeper message.

- The comedy was either unnecessarily hacky, or just plain crude, calculated to create shocked humor, without leading up to a deeper plot.

- Most comedy is flat or hack-ish or just plain crude, calculated to evoke shocked laughs, without opening up to a deeper audience.

(a) ACS

- This comedy is either plain hacky, or just plain crude, calculated to provoke shocked discussion, without linking up to a deeper message.

- And comedy that can be hacky, or just plain crude, calculated to provoke shocked discussion, without opening up to a deeper topic.

- Modern comedy can be deliberately hacklish, or just plain crude, calculated to provoke shocked disbelief, without opening up to a deeper meaning.

- Simple comedy has all things hacky, or just plain crude, calculated to be shocked away, without opening up to a deeper meaning.

- A comedy usually ranges from hacky, or just plain crude, calculated to provoke shocked surprise, without opening up to a deeper reality.

(b) DMALA

Figure 13: Text Generations using ACS and DMALA. As demonstrated empirically in 2, the ACS examples demonstrate higher diversity than the DMALA generations.

