# OpenReview forum: "Gradient-based Discrete Sampling with Automatic Cyclical Scheduling"
_NeurIPS.cc/2024/Conference — NeurIPS 2024 poster_

### Official Review · Reviewer_V8nn · 2024-07-11

**Soundness:** 3
**Presentation:** 2
**Contribution:** 3
**Rating:** 6
**Confidence:** 2

**Summary:**

The paper presents a novel gradient-based algorithm to sample from complex multimodal discrete distributions based on differentiable energy functions. Overall, the method is based on locally balanced proposals, previously introduced, and instantiates it with parametrized functions and a cyclical schedule for the "learning rate" that promotes the alternation of modes discovery and modes refinement.
The authors also provide an algorithm to automatically tune the introduced parameters based on an input acceptance rate and initial and final balancing parameters.
The paper features a theoretical analysis that includes concrete convergence rates under (somewhat restrictive) assumptions and finishes with some experiments in learning and sampling from RBMs and EBMs and finally in text infilling with masked language models.

**Strengths:**

- to the best of my (limited) knowledge in this area, the algorithm presented in the paper seem novel and potentially quite impactful (provided the author release an "easy-to-use" implementation)
- the illustrative example sets the stage nicely for the need of further development in the field of gradient-based sampling from discrete distribution and provides a very compelling visualization of the efficacy of the method.
- the non-asymptotic bounds may offer concrete guarantees when assumptions are met.

**Weaknesses:**

- I think too much of the paper is in the appendix, and the information presented in the paper is not sufficient to fully follow the logical flow (see also points below). In my opinion, some details regarding the development of the method and the theory could be moved to the appendix to make more space for both preliminaries (like how are these methods used in practice) and developing more intuition.
- the algorithm is fairly complex and the paper fails in provide enough intuition for some parts of its functioning. Like the authors claim that "it is fairly easy to choose initial and final balancing factors", but do not elaborate why (in the main paper)
- the only "real-world" experiment is only sketched in the main paper, not providing enough information to appreciate the task. What is the precise difficulty here? Since roberta is a masked language model I believe it is possible to derive "pseudo-distributions" like in [1]
- the theoretical analysis seems to require strong assumptions that may be violated in compelling real-world use cases (like LLMs)


[1] Hennigen, Lucas Torroba, and Yoon Kim. "Deriving language models from masked language models." arXiv preprint arXiv:2305.15501 (2023).

**Questions:**

- Can you please discuss in which cases the locally concave hypothesis holds for realistic models such as LLMs?
- Even if this is probably more of a sanity check, I'd like to see how the method behave on unimodal, or mildly multimodal distributions
- Can you please describe some other applications of the method to real-world problems (like text infilling)

**Limitations:**

Partially discussed, see questions

---

> ### Author Rebuttal · Authors · 2024-08-07
>
> Thank you for your supportive comments. We include our responses to your points below:
>
> # Q1: Insufficient Content in Main Body regarding Tuning Algorithm
> In section 4.4, we first provide an intuition for our algorithm under “Main idea” and then present our algorithm by separating it into three separate components: the estimation of $\alpha_\text{max}$, the estimation of $\alpha_\text{min}$, and constructing the schedule of $\beta_i$. Section A of the appendix provides the detailed algorithm that shows how each sub-routine works. We will move some content from the Appendix to the main body and try our best to further improve its clarity.
> # Q2: Choosing $\beta_\text{max}, \beta_\text{min}$
> In section 4.3 where we introduce the cyclical balancing parameter schedule in lines 139-146, we have discussed that the selection of these values are dependent on the theoretical results from [1], [2], which demonstrate that when the step-size $\alpha \to 0$, the optimal $\beta_\text{min} = .5$; but when $\alpha \to \infty$, the optimal $\beta_\text{max} = 1$. We will move the explanation of this from the Appendix to the main body.
> # Q3: Difficulty of Infilling Task
> The difficulty of the text infilling task comes from the fact that the probability distribution is over a very large sample space due to the size of the vocabulary and the number of positions to be filled, as discussed in [3]. Furthermore, the paper you mention [4] only considers how to find the distribution of two masked tokens, whereas we mask up to 50% of the sentence. This greatly increases the complexity of the task as it exponentially increases the sample space of potential combinations.
> # Q4: Locally Concave Hypothesis in Practice
> The locally log-concave assumption is common in both sampling and optimization literature [4], [5]. It holds on several practical discrete sampling tasks, such as Ising Models with negative definite weight matrix $W$ and Poisson distributions.
> The locally log-concave assumption does not hold with complex models such as LLMs, since theoretical results of sampling on such models are in general difficult to obtain. Our work provides the first non-asymptotic convergence bound for gradient-based discrete samplers. We leave analysis on non log-concave distributions for future work.
> Besides, we provide extensive empirical work demonstrating that our sampler converges on models where this assumption does not hold, such as deep EBMs and LLMs.
> # Q5: Unimodal, mildly multimodal distributions
> We demonstrate the performance of ACS on a unimodal distribution, similar to the experiment from Figure 1. We put the links for the visual results below.
>
> Target Distribution: https://anonymous.4open.science/r/neurips_rebuttal-B010/single_mode_init_dist.pdf \
> DMALA Estimated Distribution: https://anonymous.4open.science/r/neurips_rebuttal-B010/single_mode_est_dist_dmala.pdf \
> ACS Estimated Distribution: https://anonymous.4open.science/r/neurips_rebuttal-B010/single_mode_est_dist_acs.pdf
>
> Below we provide the results for DMALA and ACS on a mildly multimodal distribution, where the majority of the mass is placed on one mode.
>
> Target Distribution: https://anonymous.4open.science/r/neurips_rebuttal-B010/slightly_multimodal_target_dist.pdf \
> DMALA Estimated Distribution: https://anonymous.4open.science/r/neurips_rebuttal-B010/slightly_multimodal_dmala_est_dist.pdf \
> ACS Estimated Distribution: https://anonymous.4open.science/r/neurips_rebuttal-B010/slightly_multimodal_acs_est_dist.pdf
>
> We provide quantitative results below, where we compare the KL Divergence between the estimated and true distribution.
> | Distribution              | DMALA |    ACS |
> | :-| :-: | :-: |
> | Slightly Multimodal |   $0.7011$   | $0.1250$ |
> | Unimodal  |   $0.0089$   | $0.0032$ |
>
> In both cases, we see ACS achieves a lower KL divergence than DMALA, thus demonstrating that our proposed method is capable of capturing both a unimodal and slightly multimodal distribution.
> # Q6: Other applications of method
> In addition to sampling from language models, we also demonstrate that our proposed sampler can be used to train deep discrete energy based models more efficiently than previous discrete samplers.
>
> Within the domain of language modeling, there are many additional applications beyond text infilling. In work such as [7], [8], language models are framed as Energy Based Models and then sampled from in order to perform controlled generation tasks. These tasks include abductive reasoning and counterfactual story rewriting [8]; sentiment guided generation and detoxification as in [7]; and keyword generation as in [7], [8]. Given the success of applying ACS and discrete sampling to the task of text infilling, a natural step would be to investigate the application of ACS to controlled language generation.
>
> Furthermore, controlled generation is remarkably similar to language model alignment as described in [9]. One interesting direction would be to apply the discrete Energy Based Model framework to this task as a decoding time algorithm, similar to [10].
>
> [1] Informed proposals for local MCMC in discrete spaces. 2017.\
> [2] Any-scale Balanced Samplers for Discrete Space. ICLR 2022.\
> [3]. A Langevin-like Sampler for Discrete Distributions. ICML 2022.\
> [4] Deriving language models from masked language models. arXiv Preprint 2023.\
> [5] Optimization methods for large-scale machine learning. SIAM Review 2018.\
> [6]  Theoretical guarantees for approximate sampling from smooth and log-concave densities. Journal of the Royal Statistical Society Series B: Statistical Methodology 2017.\
> [7] Gradient-Based Constrained Sampling from Language Models. EMNLP 2022.\
> [8] COLD Decoding: Energy-based Constrained Text Generation with Langevin Dynamic. NeurIPS 2022.\
> [9] Reward-augmented decoding: Efficient controlled text generation with a unidirectional reward model. EMNLP 2023.\
> [10] Args: Alignment as reward-guided search. ICLR 2024.

---

> > ### Comment · Reviewer_V8nn · 2024-08-09
> > **Thanks**
> >
> > Dear authors. Thank you very much for your rebuttal. I appreciate the additional experiments clarifications on the infilling task. I keep my opinion that this work is a valid addition to the conference.

---

### Official Review · Reviewer_iri7 · 2024-07-12

**Soundness:** 3
**Presentation:** 3
**Contribution:** 3
**Rating:** 6
**Confidence:** 4

**Summary:**

The paper proposes a solution to the challenge of sampling from high-dimensional discrete spaces, where conventional discrete samplers often get trapped in local modes. To address this, the authors introduce a discrete Langevin sampler with automatic cyclical scheduling. This method comprises three components: a cyclical step size schedule, a cyclical balancing schedule, and an automatic hyperparameter tuning scheme. The authors provide theoretical guarantees for non-asymptotic convergence and inference, and extensive experiments demonstrate the method's superiority in sampling complex multimodal discrete distributions.

**Strengths:**

The paper is well-motivated, and the proposed automatic cyclical scheduling method is presented clearly, making it accessible to readers. The theoretical results, which offer non-asymptotic convergence, support the method's robustness. Additionally, the empirical study is solid, with extensive experiments demonstrating the method's superiority in sampling from high-dimensional spaces.

**Weaknesses:**

- My primary concern lies in the complexity of the proposed methods. The automatic schedule tuning scheme appears to be quite time-consuming, particularly the grid search required for the balancing parameters \beta_i, demanding significant computational resources.
- Another concern pertains to the theoretical assumptions underlying the analysis. The non-asymptotic convergence of the proposed samplers relies on the strong convexity of the negative energy function, an assumption that may not hold in practical deep EBM scenarios. Despite this potentially restrictive condition, the analysis provides valuable insights, and empirically, the proposed method demonstrates effective performance, as supported by extensive studies.

**Questions:**

- In equations 8 and 9, how do you estimate the acceptance rate A? Is it estimated by averaging across the training batch? If so, that implies the complexity would increase to n*s times compared to the original DMALA samplers at each step, where n is the number of grids for parameters \beta_i and s is the number of sampling steps per cycle. This would be highly time-consuming.
- Could you provide a comparison of the running times between the proposed ACS sampler and the DMALA samplers?
- Could you elaborate on the rationale behind setting the target acceptance rate $\rho^*$ to 0.5 in your experiments? What are the implications of setting it to 1 or 0.234 instead?

**Limitations:**

Despite the strong assumptions in the theoretical analysis and the perceived complexity of the algorithm, this paper presents a substantial contribution to the field by offering a well-explained, theoretically sound, and empirically validated approach to enhancing discrete sampling in high-dimensional spaces.

---

> ### Author Rebuttal · Authors · 2024-08-07
>
> Thank you for your supportive and valuable comments. We will address the issues you raise below.
> # Q1: Complexity of Tuning Algorithm
> In total, the automatic tuning algorithm takes 500 sampling steps as a budget at maximum, which is much smaller when compared to the 5,000 sampling steps we use for EBM sampling and the 10,000 steps we use for RBM sampling. Furthermore, the cost of a tuning step is almost the same as a standard sampling step — as shown in Algorithm 4 and Algorithm 5, most of the additional steps for the estimation of $\alpha_\text{min}, \alpha_\text{max}$, and the $\beta_i$ schedule are arithmetic operations that take constant time or averaging over acceptance rates that are already computed within a normal sampling step. Therefore, the tuning algorithm is neither time-consuming nor costly compared to the main sampling costs.
> # Q2: Estimation of Acceptance rate
>
> The acceptance rate for a pair of $\alpha, \beta$ is calculated by averaging over the current batch for one iteration. As most MCMC algorithms run multiple chains at the same time by sampling in batches, we found that averaging over the batch for a single time step provides useful acceptance rates to use during the tuning process. For a given time step, we take the average acceptance probability as calculated by Equation 6 across the batch. Thus each tested pair of $\alpha, \beta$ requires only one step to estimate the acceptance rate.
>
> While this does mean that the complexity is $O(n * s)$, where $n$ is the number of potential $\beta_i$ and $s$ is the number of steps per cycle, we found that the number of potential $\beta_i$ does not have to be very large. In practice, we found that testing 5 different $\beta_i$ for each step produces good schedules. The largest cycle length that we use in our experiments is 20 steps — this means that this step takes 100 sampling steps total, which is small compared to the total number of sampling steps of 10,000 in the RBM sampling experiment and 5,000 in the EBM sampling experiment.
> # Q3: Run Time Comparison
> We found that using a budget of 500 sampling steps enabled the discovery of good hyper-parameter schedules. As discussed in our response to Q2, a tuning step has essentially the same cost as a sampling step. Thus the total overhead amounts to 5% of the total budget of the RBM sampling task and 10% of the total budget of the EBM task. Beyond this overhead, the run times of DMALA and ACS are the same, as both calculate the proposal function and the acceptance rate in the exact same manner.
>
> We use the RBM sampling experiment to further compare the run times between DMALA and ACS. We we make the results available at the following link: https://anonymous.4open.science/r/neurips_rebuttal-B010/rbm_log_mmds_dmala_acs_comp_time.pdf
>
> As visible in the results, even if we restrict ACS (including the tuning phase) and DMALA to the same total time budget of 20 seconds, we see that ACS is able to achieve a log-MMDs superior to that of DMALA. Thus even considering the overhead, our proposed sampler outperforms DMALA within a fixed time budget.
>
> # Q4: Rationale behind setting target acceptance rate
> We base our target acceptance rate of .5 on the study done in [1], where the authors demonstrate that the ideal acceptance rate for locally balanced samplers is close to .5. Besides, .5 is also the commonly used target acceptance rate for gradient-based samplers. Therefore, we use .5 in the paper and find that it works well in practice. If we set the target acceptance rate to be close to 1, we will end up with step-sizes that are too small, resulting in insufficient exploration of the distribution. If we set the target acceptance rate to be around .234, then the sampler will end up rejecting most of the proposed movement, decreasing the efficiency of the sampler. An acceptance rate of .5 avoids either scenario, allowing for efficient and thorough characterization of the target distribution.
>
> [1]. Optimal Scaling for Locally Balanced Proposals in Discrete Spaces. NeurIPS 2022.

---

> > ### Comment · Reviewer_iri7 · 2024-08-12
> >
> > Thanks for the authors' response. It has solved my concern about the complexity. I keep my opinion that this work is good to be in, and I highly recommend including the discussion of complexity in the camera ready.

---

### Official Review · Reviewer_1xVN · 2024-07-12

**Soundness:** 3
**Presentation:** 3
**Contribution:** 2
**Rating:** 5
**Confidence:** 1

**Summary:**

The paper introduces a novel method for sampling from multimodal discrete distributions, which presents an innovative approach to address the challenge of local modes trapping in gradient-based discrete sampling, together with non-asymptotic convergence guarantee and empirical validation of the proposed method.

**Strengths:**

1. The proposed method seems novel to address the challenge of sampling from multimodal discrete distributions.
2. The hyperparameter tuning algorithm seems useful for practical use.

**Weaknesses:**

1. In the experiments, there seems to be no error bars in Figure 1 and Table 1.
2. The quality of the samples seems worse than DMALA in Table 2 and Figure 12. Is it possible that the proposed method might sacrifice sample quality to achieve higher diversity?

**Questions:**

1. The paper mentions the proof is not consistent with the specific tuning algorithm used in the experiments. Could you elaborate on the reasons?

---

> ### Author Rebuttal · Authors · 2024-08-07
>
> Thank you for your insightful comments. We answer the questions below.
>
> # Q1: Error Bars Fig 1, Table 1
> Figure 1 corresponds to density estimation, where error bars would not make sense. If you are referring to Figure 3, it should be noted that the shaded area represents the range within 1 standard error of the average performance across 11 seeds for RBM sampling. For the other missing error bars, we include the updated results in the summary response, which show that our claims still hold.
>
> # Q2: Sample quality
> The proposed ACS does not sacrifice sample quality to achieve higher diversity. It should be noted that Figure 12 does not show that ACS decreases quality — the generated sentences are reasonably fluent and comparable to those generated by DMALA. Furthermore, while our method does result in higher perplexity, which corresponds to lower likelihood under the model generations, it should be noted that we include the CoLA scores to measure the grammatical quality of the generations. As indicated by these scores, our method does not sacrifice grammatical correctness for diversity.
>
> While perplexity is a popular means of evaluating language generations, it is important to recognize that perplexity is based on the likelihood of the generation under the language model. This metric is biased towards frequent patterns and does not account for diverse modes. Therefore, it does not completely align with the goal of MCMC, which is to accurately characterize the target distribution.
>
> Minimizing the average perplexity of the sample corresponds to maximizing the average likelihood of the generations, which can be at odds with the goal of accurately capturing the target distribution. We illustrate this in the following experiment, where we construct a synthetic dataset of 25 modes where the top-left mode is weighted far more than all the others. Below are the anonymized links to the images for this experiment:
>
> Target Distribution: https://anonymous.4open.science/r/neurips_rebuttal-B010/slightly_multimodal_target_dist.pdf \
> DMALA Estimated Distribution: https://anonymous.4open.science/r/neurips_rebuttal-B010/slightly_multimodal_dmala_est_dist.pdf \
> ACS Estimated Distribution: https://anonymous.4open.science/r/neurips_rebuttal-B010/slightly_multimodal_acs_est_dist.pdf
>
> | Method              | Average Energy |    KL Divergence |
> | :---------------- | :-----------: | :-----------: |
> | DMALA |   $-2.66 \pm 1.68$   | $0.70$ |
> | ACS  |   $-3.39\pm 1.63$   | $0.13$ |
>
> The average energy for the samples from DMALA is higher than that of ACS as DMALA ends up being trapped by the top left mode. This indicates that the samples generated by DMALA are more likely than the samples generated by ACS. However, the visualizations of the estimated distributions show that DMALA ignores the majority of the low-likelihood modes, whereas ACS is able to explore all of them. This is supported by the measured KL divergence between the estimated distribution and the target distribution. This demonstrates the disconnect between maximizing the average likelihood of the generated samples and accurately capturing the target distribution.
> # Q3: Consistency between Theory and Algorithm
>
> In the Conclusion and Limitations section, we mention that the proof is not based on a specific tuning algorithm. This means that our theoretical analysis does not consider the effect of the tuning algorithm. If we were to take into account the hyperparameters (alpha and beta) provided by the tuning algorithm, it might be possible to make the results more tailored to practical performance.
>
> The reason for this limitation is that conducting such a theoretical analysis is a challenging problem, as it requires analyzing an inhomogeneous Markov chain. Techniques to analyze such chains in discrete spaces, which may be applied here, are not known to us. In fact, we conjecture that any improvement on the current bound provided by us, in terms of the entire schedule, shall involve developing fundamental theory for inhomogeneous Markov chains in discrete spaces.
>
> [1]. Annealed Importance Sampling. Technical Report 1998.\
> [2]. Oops I Took A Gradient: Scalable Sampling for Discrete Distributions. ICML 2021.\
> [3]. A Langevin-like Sampler for Discrete Distributions. ICML 2022.\
> [4]. Path auxiliary proposal for MCMC in discrete space. ICLR 2022.

---

> > ### Comment · Reviewer_1xVN · 2024-08-11
> >
> > Thank you for the response and clarifications. I will keep my score.

---

### Official Review · Reviewer_1ido · 2024-07-15

**Soundness:** 3
**Presentation:** 3
**Contribution:** 2
**Rating:** 5
**Confidence:** 2

**Summary:**

This paper proposes a new discrete sampling method called ACS that addresses a common problem for existing gradient-based  approach where they are susceptible to becoming trapped in local modes. ACS combines local-balancing proposals with a cyclic step size to balance local exploitation and global exploration; it is in essence an extension of cyclic stochastic-gradient MCMC to discrete distributions.  To ensure proposals are still balanced with a varying step size, ACS uses a cyclic balancing
schedule along with an automatic tuning scheme to easily adapt the schedules.  Non-asymptotic convergence guarantees are provided.  Results demonstrate ACS to outperform prior approaches for sampling from energy based models, training RBMs, and text-infilling.

**Strengths:**

- Using a cyclic step size schedule is a well motivated and effective approach for incorporating global considerations into the original local self-balancing MCMC approach presented in https://arxiv.org/pdf/2109.03867.
- ACS is accompanied by an automated tuning scheme to make it easy to configure the two cyclic schedules.
- Strong empirical results on EBM tasks.
- ACS has non-asymptotic convergence guarantees.

**Weaknesses:**

- Experimental results for RBM have discrepancies wrt results reported in previous papers.  In particular the ranges differ for average energy from other papers and the curves for log MMD for ACS show unexpected curvature.  Please respond to the questions in the section below to clarify.
- Inadequate discussion of text-infilling results.  It is not clear why higher perplexity and diversity is good for ACS since the goal is to be able to efficiently sample from a target discrete distribution.
- Error bars missing for results in Figure 3 and Table 1.

Typos & formatting:
- Inline citation format should show author name
- Figure 3 caption: cpnvergence -> convergence

**Questions:**

- Why is average energy in Figure 3 negative?  Also, the results for ACS on dynamic_mnist and omniglot are unexpected with better performance on fewer iterations before converging.
- Why is the scale for average energy in Figure 3 different from that reported in Figure 4 of the [AB paper](https://openreview.net/pdf?id=lEkl0jdSb7B)?

**Limitations:**

Limitations are adequately discussed.

---

> ### Author Rebuttal · Authors · 2024-08-07
>
> Thank you for the thoughtful review. We answer your questions below.
> # Q1: Missing Error Bars, Figure 3 and Table 1
> Thank you for pointing this out. For the log MMD curves for the RBM experiments in Figure 3, it should be noted that the filled in area corresponds to values within one standard error of the mean across 11 different random seeds. However, it is correct that our curves for the mixing time for deep energy based models in Figure 3 and the metrics from Table 1 do not have standard error bars. We have provided the results with error bars in our summary response, which show that our claims still hold.
> # Q2: Negative Average Energy
> The average energy is negative due to the definition of the distribution we use. We define the target distribution as $\pi(\theta) = \frac{\exp U(\theta)}{Z}$, where $Z$ is the partition function. The energy is $U(\theta) = log (\pi (\theta)) - log Z$. Because a log probability is negative and the partition function is the sum of exponentials, which are positive, it should be the case that the energy function is negative. In contrast, [5] use the definition of the distribution from [4] for the experiments with deep energy based models as they train these models using the sampling algorithm proposed in [4]. They define the distribution as $\pi(\theta) = \frac{\exp (-U(\theta))}{Z}$. Thus, their energy is positive.
> # Q3: Better performance with fewer iterations before converging
> Here, it should be noted that higher energy of the batch does not correspond to closeness to the target distribution. For a more in depth explanation on the difference between higher average energy and closeness to target distribution, we include a toy example in our response to Q5.
>
> The energy decrease can be seen as the direct result of ACS being able to escape from modes quicker than other samplers — ACS is able to find very likely samples quickly, but then explores the different modes of the distribution, causing the energy to decrease.
>
> Because the ground truth distribution in deep EBMs is unknown, we evaluate the sampling performance using average energy following the experiment in [5]. It tells us how quickly different samplers are able to reach likely modes, which gives insight into the **speed** of the various samplers. To evaluate the convergence to the target distribution, we include extensive experiments on RBMs, where it is possible to get an estimate as to how close the sampler is to the ground truth distribution via the maximum mean discrepancy to the block gibbs sampler, which takes advantage of the known architecture of an RBM. This enables us to get insight as to how **accurate** the samplers are.
> # Q4: Average Scale of energy different than AB
> This is due to the training of the EBMs — because the EBM represents an unnormalized probability distribution, different EBMs tend to have different scales of energy, depending on the sampler used within the contrastive learning routine. [5] uses Path Auxiliary MCMC from [4], whereas we use gibbs with gradient for the models in our EBM experiment.
> # Q5: Higher perplexity and diversity in text infilling task
> While we include perplexity as it is a popular means of measuring language quality, this metric faces significant limitations as discussed in the literature (e.g.[6,7]). It fails to capture logical or grammatical coherence, bias towards frequent patterns, does not align with human evaluation, and cannot capture diversity. Because of this, we further include CoLA (measure the grammatical quality) and Self-Bleu (measure diversity) to comprehensively evaluate the generated sentences. The results show that ACS generations achieve better CoLA and Self-Bleu scores.
>
> Furthermore, perplexity is especially limited when trying to measure the accuracy of a sampler in regards to a target distribution. As perplexity is based on the likelihood of the generation under the language model, it does not account for diverse modes. To illustrate this point, we provide the following toy example to show the goals of obtaining the most likely generations and capturing the language model distribution are orthogonal. Similar to Figure 1, we construct a synthetic distribution of 25 modes where 1 mode is weighted heavier than the others. We provide visualizations along with a quantitative comparison of the estimated and target distributions below.
>
> Target Distribution: https://anonymous.4open.science/r/neurips_rebuttal-B010/slightly_multimodal_target_dist.pdf \
> DMALA Estimated Distribution: https://anonymous.4open.science/r/neurips_rebuttal-B010/slightly_multimodal_dmala_est_dist.pdf \
> ACS Estimated Distribution: https://anonymous.4open.science/r/neurips_rebuttal-B010/slightly_multimodal_acs_est_dist.pdf
> | Method              | Average Energy |    KL Divergence |
> | :---------------- | :-----------: | :-----------: |
> | DMALA |   $-2.66 \pm 1.68$   | $0.70$ |
> | ACS  |   $-3.39\pm 1.63$   | $0.13$ |
>
> During the sampling process, DMALA becomes stuck at the high likelihood mode, preventing the sampler from exploring the rest of the sample space. In contrast, ACS is able to visit all the modes in the distribution. This can be seen through a visual inspection of the density maps for DMALA and ACS. While DMALA ends up generating samples with higher energy, ACS estimates a distribution that is far closer to the ground truth as measured by the KL divergence. This demonstrates that generating more likely samples does not correspond to accuracy in terms of convergence to the target distribution.
>
> [1]. Annealed Importance Sampling. Technical Report 1998.\
> [2]. Oops I Took A Gradient: Scalable Sampling for Discrete Distributions. ICML 2021.\
> [3]. A Langevin-like Sampler for Discrete Distributions. ICML 2022.\
> [4]. Path auxiliary proposal for MCMC in discrete space. ICLR 2022.\
> [5]. Any-scale Balanced Samplers for Discrete Space. ICLR 2023.\
> [6]. Lower Perplexity is Not Always Human-Like. ACL 2021\
> [7]. Language model evaluation beyond perplexity. ACL 2021

---

> > ### Comment · Reviewer_1ido · 2024-08-13
> > **Post author response**
> >
> > Thank you for responding to my questions.  I have changed my score from 4 to 5.

---

### Author Rebuttal · Authors · 2024-08-07

We would like to thank all the reviewers for their constructive reviews. Multiple reviewers have pointed out that our results for sampling from Energy Based Models in Figure 3 and the Annealed Importance Sampling results in Table 1 do not have error bars. Below, we have provided the updated results for Table 1 and Figure 3, both of which confirm that our observations and conclusions remain valid.


Updated Figure 3 EBM Mixing Results: https://anonymous.4open.science/r/neurips_rebuttal-B010/ebm_sample_avg_iter.pdf

Updated Table 1:
| Dataset              | DMALA |    ACS |
| :---------------- | :---------: | :---------: |
| Static MNIST        |   $-80.031 \pm 0.038$   | $-79.905 \pm 0.057$ |
| Dynamic MNIST           |   $-80.120 \pm 0.036$   | $-79.634 \pm 0.024$ |
| Omniglot    |  $-99.243 \pm ​​2.101$   | $-91.487 \pm 0.128$ |
| Caltech |  $-98.001 \pm 0.371$   | $-89.262 \pm 0.290$ |


In the updated Table 1, we ran the AIS evaluation for the models trained with various samplers over 8 random seeds, and we show the mean as well as the standard error for the log likelihood over the test set. The results indicate that our proposed sampler is capable of training the models of better quality given the same computational budget.

In the updated Figure 3, we show the area within one standard error of the mean across 11 different random seeds for each time step. Our proposed ACS has consistent performance across the datasets in terms of being able to mix in quickly. On Static MNIST, Dynamic MNIST, and Omniglot, we observe that our proposed ACS is able to consistently find high energy modes far quicker than the other methods. ACS is then able to find the less likely modes due to its ability to escape from local modes. On Caltech Silhouettes, ACS converges faster at the beginning and maintains competitive performance against the baselines.

Finally, we would like to emphasize the key contributions of this work. There has been much work regarding the use of gradient-based discrete samplers, but not enough investigation as to how these gradient-based samplers are affected by the multi-modal nature of high dimensional discrete distributions. We demonstrate that this is a current limitation of discrete gradient-based samplers, and introduce a method that is capable of avoiding this pitfall while retaining the benefits associated with gradient information. Our work introduces cyclical step size and balancing parameter schedules with theoretical guarantees. Furthermore, our method can be automatically configured by a novel tuning algorithm with minimal overhead. We demonstrate on a synthetic highly multimodal distribution and a range of datasets that our sampler can achieve superior performance over existing methods. Given both the theoretical contributions and experimental results for our method, we believe this is a valuable contribution to the field of discrete MCMC sampling.

---

### Decision · Program_Chairs · 2024-09-25

**Decision:**

Accept (poster)

**Comment:**

This paper proposes an automatic cyclical scheduling, designed for efficient and accurate sampling in multimodal discrete distributions. The authors prove a non-asymptotic convergence guarantee in general discrete distributions. The method is also verified by experiments.

All reviewers agree to accept this paper because (1) using a cyclic step size schedule is well motivated, (2) an automated tuning scheme is used, making it easy to configure the two cyclic schedules, (3) strong empirical results, (4) non-asymptotic convergence guarantees, etc. I agree with the opinions and recommend an accept.